# Divergent WNT signaling and drug sensitivity profiles within hepatoblastoma tumors and organoids

Thomas A. Kluiver [1,11], Yuyan Lu [1,2,11], Stephanie A. Schubert [1], Lianne J. Kraaier[1], Femke Ringnalda [1], Philip Lijnzaad [1], Jeff DeMartino [1,3], Wouter L. Megchelenbrink [1,4], Vicky Amo-Addae[1], Selma Eising[1], Flavia W. de Faria[5], Daniel Münter[5], Marc van de Wetering [1], Kornelius Kerl [5], Evelien Duiker [6], Marius C. van den Heuvel[6], Vincent E. de Meijer [7], Ruben H. de Kleine [7], Jan J. Molenaar[1], Thanasis Margaritis [1], Hendrik G. Stunnenberg [1], Ronald R. de Krijger[1,8], József Zsiros[1], Hans Clevers [1,3,9,10] & Weng Chuan Peng [1] ✉

Hepatoblastoma, the most prevalent pediatric liver cancer, almost always carries a WNT-activating *CTNNB1* mutation, yet exhibits notable molecular heterogeneity. To characterize this heterogeneity and identify novel targeted therapies, we perform comprehensive analysis of hepatoblastomas and tumor-derived organoids using single-cell RNA-seq/ATAC-seq, spatial transcriptomics, and high-throughput drug profiling. We identify two distinct tumor epithelial signatures: hepatic 'fetal' and WNT-high 'embryonal', displaying divergent WNT signaling patterns. The fetal group is enriched for liver-specific WNT targets, while the embryonal group is enriched in canonical WNT target genes. Gene regulatory network analysis reveals enrichment of regulons related to hepatic functions such as bile acid, lipid and xenobiotic metabolism in the fetal subtype but not in the embryonal subtype. In addition, the dichotomous expression pattern of the transcription factors HNF4A and LEF1 allows for a clear distinction between the fetal and embryonal tumor cells. We also perform high-throughput drug screening using patient-derived tumor organoids and identify sensitivity to HDAC inhibitors. Intriguingly, embryonal and fetal tumor organoids are sensitive to FGFR and EGFR inhibitors, respectively, indicating a dependency on EGF/FGF signaling in hepatoblastoma tumorigenesis. In summary, our data uncover the molecular and drug sensitivity landscapes of hepatoblastoma and pave the way for the development of targeted therapies.

Hepatoblastoma, the most prevalent liver malignancy in children, is slowly increasing in incidence[1–3]. Although cisplatin-based chemotherapy regimens and surgical resection have improved survival[4–6] for patients with low-risk tumors, the prognosis of children with advanced or high-risk hepatoblastoma (60% of all patients) remains unsatisfactory due to a high recurrence rate and progressive disease stemming from the development of chemoresistance[7]. Moreover, long-term side effects from chemotherapy, such as hearing loss, reduced cardiac and renal function, and secondary cancers, can severely impact childhood development and quality of life

in survivors[8]. Improvements in care for these patients require the development of targeted therapeutic treatments.

Hepatoblastomas have a low mutational burden and few recurrent chromosomal aberrations[9,10]. Notably, they are characterized by the presence of activating mutations in *CTNNB1*, which encodes β-catenin, or in other WNT pathway genes, accounting for over 90% of cases[2]. These tumors are believed to originate from the aberrant expansion of hepatic progenitors that harbor *CTNNB1* mutations during fetal liver development and morphologically resemble stages of liver development[11]. Within the tumor epithelium, two main histological types are commonly observed: the more differentiated 'fetal' histology, which resembles the fetal liver developmental stage, and the less differentiated 'embryonal' histology, resembling an earlier stage of liver development. Often, tumors contain both fetal and embryonal histological components. It is generally hypothesized that tumors enriched in embryonal components are associated with worse prognosis[12].

The molecular profile of hepatoblastoma has been characterized previously by several groups using bulk transcriptomic methods[13–18]. These studies, based on different cohorts, generally agree on the existence of three main molecular subgroups within hepatoblastoma, i.e., tumors with a predominantly differentiated fetal histology, enriched in a hepatic signature, tumors with a predominant embryonal histology, enriched in a progenitor signature, and a third group characterized by a mesenchymal signature[14–20]. More recently, single-cell transcriptomic studies based on a small cohort of patients reported a classification in five[21] or seven tumor cell clusters[22]. In addition, several studies have reported biological markers that showed a correlation with clinical behavior and outcome, such as the 16-gene signature[14], four-gene signature[19], expression of vimentin, and the 14q32-gene signature[20]. However, the transcriptomic heterogeneity observed in hepatoblastoma, especially at the single-cell level, remains largely unexplored.

In this study, we utilize single-cell RNA-sequencing (scRNA-seq), single-cell assay for transposase-accessible chromatin with sequencing (scATAC-seq), spatial transcriptomics (ST), and gene regulatory network analysis to investigate the molecular landscape of hepatoblastoma (Supplementary Fig. 1a). Focusing on the tumor epithelial component, we describe two distinct tumor signatures corresponding to the fetal and embryonal components of hepatoblastoma. In addition, we establish a cohort of hepatoblastoma tumor organoids from patient tumor material, which recapitulates tumor heterogeneity and facilitates in-depth molecular characterization. These organoids allow for high-throughput drug screening and reveal signaling pathways essential for tumorigenesis. We identify several classes of inhibitors that could serve as therapeutic drugs, most notably HDAC inhibitors. Tumors with low mutational burden, such as hepatoblastoma, present a unique challenge in targeted treatment. To our knowledge, this study represents the first extensive screening effort using a large cohort of well-characterized, patient-derived hepatoblastoma organoid models across clinical stages and provides essential insights into targeted therapy for children with liver tumors.

## Results
### scRNA-seq confirmed the presence of fetal and embryonal tumor signatures in hepatoblastoma
To investigate the heterogeneity of hepatoblastoma tumor cells, we analyzed a recently published scRNA-seq dataset of nine hepatoblastomas from Song et al.[21] (Supplementary Fig. 1a, b). We focused on the epithelial tumor cells, while the mesenchymal component and the very rare tumor cluster with neuroendocrine features were excluded from our analysis. In addition, we included hepatocytes and cholangiocytes from the paired normal tissue for comparison. Single-cell data was processed as described in Experimental Procedures. In total, 1562 cells from 9 patients were jointly analyzed. Based on

unsupervised graph-based clustering, normal hepatocytes and cholangiocytes from multiple patients were present as two separate clusters (Fig. 1a). From tumor tissues, we identified two distinct clusters of cells originating from multiple patients, denoted here as 'fetal' (F) and 'embryonal' (E).

The fetal tumor cells exhibited high levels of hepatocyte markers (e.g., *ALB*), and metabolic markers in the pericentral zone typically activated by WNT signaling (e.g., *GLUL, CYP2E1,* and *RHBG)*[23] with low levels or absence of various periportal hepatic markers (e.g., *ALDOB, PCK1,* and *FBP1*) compared to the normal hepatocytes (Fig. 1b, c and Supplementary Fig. 1c). These cells also expressed general WNT target genes, such as *NOTUM* and *NKD1*, but at a lower level than the embryonal tumor cluster. In addition, these tumor cells upregulated fetal liver markers (e.g., *SPINK1, GPC3, REG3A,* and *RELN*)[14,24–27], and expressed multiple imprinted genes, including *IGF2, PEG3, PEG10, DLK1,* and *MEG3* (Fig. 1b and Supplementary Fig. 1c), with *DLK1* and *MEG3* being located on the 14q32 locus, as previously described by Cairo et al.[14] and Carrillo-Reixach et al.[20].

The embryonal tumor cluster showed significant upregulation of key WNT signaling target genes and regulators, including *APCDD1*[28], *NKD1, NOTUM, KREMEN1, TNFRSF19,* and *AXIN2*, as well as WNT target genes that are associated with epithelial-to-mesenchymal transition (EMT) such as *TWIST1, BMP4* and *VIM*[29], along with low levels of hepatic markers compared to fetal tumors and normal hepatocytes (Fig. 1b, c and Supplementary Fig. 1c). Next, using gene set enrichment analysis, we compared fetal and embryonal tumor cells with each other (Fig. 1d). Fetal tumor cells were enriched for hepatic functions, such as xenobiotic and bile acid metabolism, coagulation, and the complement system. In turn, the embryonal cells were enriched for EMT, WNT/β-catenin signaling, the p53 pathway, and mitotic spindle-related genes (Fig. 1d).

In addition, we identified the expression of fibroblast growth factor receptor 1 (*FGFR1*) and several FGF ligands in the embryonal tumor cluster. In the fetal tumor cluster, we observed expression of epidermal growth factor receptor (*EGFR*), also present in normal hepatocytes and cholangiocytes (Fig. 1c).

To validate the presence of fetal and embryonal tumor signatures in other patient cohorts, we performed additional scRNA-seq analysis on two tumor samples from the Princess Máxima Center (PMC) and a previously published single nucleus RNA sequencing (snRNA-seq) dataset (Hirsch et al., Cancer Discovery, 2021)[17]. We identified both fetal and embryonal tumor cells across the three samples (Fig. 1e, f). Indeed, the subtype-specific markers in these samples showed extensive overlap with our described signatures (Fig. 1g and Supplementary Fig. 1d).

Our fetal tumor signature also shows extensive overlap in genes from the Hepatoblast I and II signatures described in Song et al.[21] (Supplementary Fig. 1e, f). A distinct embryonal signature was not identified in the original analysis by Song et al.[21] (Supplementary Fig. 1e). However, enrichment of our embryonal tumor genes was observed in the neuroendocrine and DCN-high tumor signatures, with some overlapping WNT markers in neuroendocrine and mesenchymal markers in DCN-high tumor cells (Supplementary Fig. 1e).

Next, we compared our signatures to previously described 'hepatic' and 'progenitor' hepatoblastoma tumor signatures, identified by Cairo et al.[14], Hirsch et al.[17], and Nagae et al.[18] using bulk transcriptomic profiling. Notably, the marker selection for the transcriptomic classifications of Cairo et al. and Hirsch et al. was limited (7-8 genes/signature) and varied between studies. To enable a more comprehensive comparison, we employed more extensive gene lists from these studies (Supplementary Data 1; 25–200 genes/signature). Our analysis revealed that the fetal tumor profile strongly correlates with previously described 'hepatic' signatures, whereas the embryonal tumor profile correlates with the 'progenitor' signatures. However, the correlation between the embryonal tumors and 'progenitor' signatures

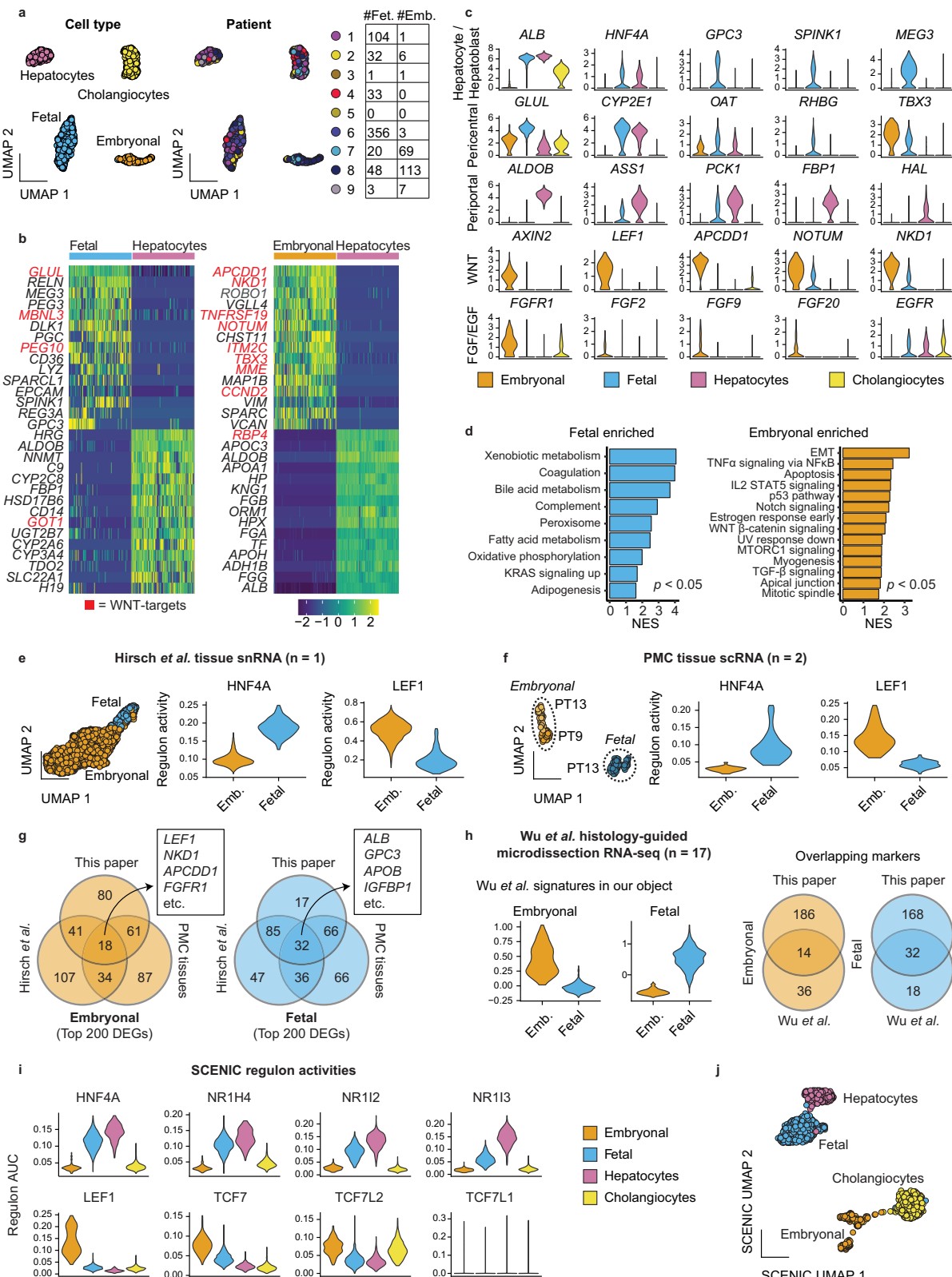

**a** Cell type / Patient UMAP plots with fetal and embryonal fractions table (#Fet. / #Emb.)

**b** Heatmaps of Fetal vs. Hepatocytes and Embryonal vs. Hepatocytes differential gene expression (= WNT-targets shown in red)

**c** Violin plots of marker genes across Embryonal, Fetal, Hepatocytes, Cholangiocytes

**d** GSEA bar plots: Fetal enriched and Embryonal enriched pathways (NES, p < 0.05)

**e** Hirsch et al. tissue snRNA (n = 1): HNF4A and LEF1 regulon activity

**f** PMC tissue scRNA (n = 2): HNF4A and LEF1 regulon activity

**g** Venn diagrams of Top 200 DEGs overlap (Embryonal and Fetal)

**h** Wu et al. histology-guided microdissection RNA-seq (n = 17): signatures and overlapping markers

**i** SCENIC regulon activities (HNF4A, NR1H4, NR1I2, NR1I3, LEF1, TCF7, TCF7L2, TCF7L1)

**j** SCENIC UMAP

across different studies was generally weaker compared to the correlations observed within the hepatic tumor signatures (Supplementary Fig. 1e–h). The actual overlap in genes between our signatures and those from other studies was limited (Supplementary Fig. 1i). Notably, our embryonal tumor signature exhibited a significant enrichment of general WNT-related genes compared to the other 'progenitor' signatures (Supplementary Fig. 1j)[14,17,18].

We further compared our tumor signatures with a very recent study by Wu et al.[30], which performed histology-guided RNA-seq on tumor tissue sections with fetal or embryonal features, obtained by laser capture micro-dissection. Interestingly, we observed a substantial overlap between our tumor signatures and their signatures (Fig. 1h), corroborating our findings based on scRNA-seq analysis. In summary, our analysis revealed the presence of two distinct profiles in

**Fig. 1 | Single-cell RNA-seq analysis of primary tumor material illustrates fetal and embryonal tumor signatures. a** UMAP plot based on unsupervised clustering, annotated per cell type or tumor signature (left) and patient identity (right) of epithelial cell subsets from the Song et al.[21] dataset. The table shows the number of cells per signature per patient. **b** Heatmaps showing the top differentially expressed genes between tumor cell populations and hepatocytes. WNT signaling pathway target genes are upregulated, especially in the WNT-high embryonal tumor subpopulation, and marked in red. **c** Violin plots showing expression of select markers. **d** Gene set enrichment analysis showing hallmark gene sets significantly enriched in the fetal and embryonal clusters, compared to each other. NES: normalized enrichment score. Adjusted *p*-value < 0.05, calculated using the fgsea package. **e** UMAP plot showing embryonal and fetal cells identified in an external snRNA-seq dataset (Hirsch et al.[17]) from a single tumor (left). Violin plots showing TF regulon activity scores for HNF4A and LEF1 for these clusters (right). **f** UMAP plot showing embryonal and fetal cells identified in scRNA-seq data of tumors from PT9 and PT13 (left). Violin plots showing TF regulon activity scores for HNF4A and LEF1 for these clusters (right). **g** Venn diagrams showing the amount of overlap between the top 200 differentially expressed genes of the scRNA clusters described in this paper, the snRNA-seq data of Hirsch et al.[17] and scRNA-seq data of PT9 and PT13 ("PMC tissues"). **h** Violin plots showing embryonal and fetal signature scores (50 genes each) derived from histology-guided laser microdissection RNA-seq of 17 patients (Wu et al.[30]) on our tumor clusters (left). Venn diagrams show the amount of overlap between the signatures from Wu et al. (50 markers each) and ours (200 markers each) for fetal and embryonal cells. **i** Violin plots showing regulon activity scores for hepatic TFs (above) enriched in fetal tumor cells and WNT/β-catenin cofactors (below) in embryonal tumor cells. **j** UMAP plot based on the SCENIC regulon activity scores.

hepatoblastoma cells: a 'fetal' tumor profile enriched with hepatic, pericentral, and fetal liver markers and an 'embryonal' tumor profile with significant enrichment of general WNT pathway target genes and reduced hepatic markers[12–18].

## Fetal and embryonal tumor cells are enriched in hepatic and WNT pathway-related regulons, respectively

To uncover the gene regulatory network underlying the different hepatoblastoma subpopulations, we employed single-cell regulatory network inference and clustering (SCENIC)[31] analysis (Supplementary Fig. 1k–m). SCENIC utilizes scRNA-seq co-expression patterns in combination with *cis*-regulatory motif analysis to infer the activity of transcription factors (TFs) and their target genes (termed 'regulon activity'). Gene regulatory network analysis has the potential to better distinguish cellular heterogeneity than RNA-seq alone, as it identifies key transcriptional activity underlying a specific cellular state[31].

Consistent with previous studies[32], our analysis revealed that cholangiocytes were enriched for regulons such as ONECUT1, HNF1B, and SOX4 (Supplementary Fig. 1m). The fetal tumor cluster displayed enrichment for hepatic-specific regulons. These included hepatic nuclear factor 4 A (HNF4A), FOXA3 (i.e., HNF3G), androgen receptor (AR), and multiple nuclear receptor subfamily members such as NR1H4 (i.e., farnesoid X-activated receptor [FXR]), NR1I3 (i.e., constitutive androstane receptor [CAR]) and NR1I2 (i.e., pregnane X receptor [PXR])) (Fig. 1e, f, i and Supplementary Fig. 1k, m). Collectively, these TFs regulate essential hepatic functions such as bile acid, lipid, and xenobiotic metabolism, and were also present in normal hepatocytes (Fig. 1d and Supplementary Fig. 1k, m). Notably, some of these TFs, such as CAR and PXR, are activated by the WNT pathway[33,34].

In contrast, the embryonal tumor cluster mostly lacked these hepatic TFs, or they were active at minimal levels. This might explain the observed low levels of hepatic markers in this subgroup (Fig. 1b, c). However, this cluster showed a clear enrichment of WNT pathway regulons[35–38], such as LEF1, TCF7, TCF7L2 and MSX2 (Fig. 1e, f, i and Supplementary Fig. 1k, m), which correlates with the observed high expression of WNT target genes (Fig. 1b, c). Although the fetal tumor cluster displayed some regulon activity for TCF7 and TCF7L2, this was notably less pronounced compared to the embryonal tumor cluster.

Next, we performed unsupervised graph-based clustering using the inferred regulon activity scores and visualized this in a UMAP plot (Fig. 1j). Hepatocytes and cholangiocytes, irrespective of patient origin, formed two distinct clusters, serving as a reference for comparison. Similarly, the fetal tumor cells from different patients clustered closely together, suggesting a high degree of similarity. In contrast, the embryonal cells formed separate clusters that were associated with specific patient origin, suggesting the presence of inter-patient heterogeneity (Supplementary Fig. 1c, k, l). Overall, our analysis uncovered distinct gene regulatory networks underlying fetal and embryonal tumor clusters and highlighted the role of hepatic-specific and WNT

pathway-related TFs in shaping the distinct tumor features observed in RNA-seq data.

## The spatial molecular landscape of hepatoblastoma illustrates tumor heterogeneity

To study the in situ expression patterns of hepatoblastoma, we performed ST analysis using the *10x Genomics Visium* platform on four hepatoblastoma tissues, as well as one paired normal liver tissue sample (Fig. 2a). The clinical information is provided in Supplementary Table 1. Using this platform, each ST spot, measured at 55 μm, contained multiple (5–50) cells. ST spots were clustered using unsupervised graph-based clustering and annotated into different regions based on marker gene expression, differential gene expression, and tissue histology.

Within the normal liver, liver metabolic zonation could readily be distinguished, with *GLUL* expression in hepatocytes adjacent to endothelial cells of the central veins, and periportal markers, such as *ALDOB*, expressed in hepatocytes adjacent to the portal triad[39,40] (Fig. 2b). As expected, in the section of distal normal liver of PT2, we could distinguish zonation patterns, including pericentral, midlobular, periportal and bile duct regions (Fig. 2c and Supplementary Fig. 2a). We observed regions with restricted *GLUL* expression and broader *CYP2E1* expression, annotated as '*GLUL* pericentral' and 'pericentral', respectively (Fig. 2d). In the periportal region, we observed expression of periportal markers, such as *ALDOB*, *HAL*, and *CYP2A7* (Fig. 2d and Supplementary Fig. 2a).

We analyzed the matched tumor tissue of PT2 (Fig. 2e and Supplementary Fig. 2b), mostly consisting of fetal hepatoblastoma (annotated as 'fetal' tumor regions) and stromal regions (containing 'stroma' and 'ductular reaction'), which were confirmed by a pathologist (R.R.d.K.). We compared the gene expression profile of the tumor cluster with the non-tumor clusters and paired distal normal tissue section (Supplementary Fig. 2c). In line with our scRNA-seq tumor signatures, the fetal tumor region upregulated fetal liver and tumor markers (e.g., *GPC3*, *SPINK1*, *REG3A*), which were not detected in the normal liver (Fig. 2e and Supplementary Fig. 2a, c, d, e). In addition, pericentral hepatic markers and WNT target genes, such as *CYP2E1* and *GLUL*, were broadly expressed in the tumor region, contrasting with the zonated pattern in the normal tissue (Fig. 2d, e). Conversely, hepatic periportal markers (e.g., *ALDOB*, *HAL*, *ASS1*) and a subset of *CYP*-related proteins (e.g., *CYP2A6*, *CYP2A7*, *CYP2B6*) were either absent or downregulated in the tumor region (Supplementary Fig. 2b, c, h).

We analyzed three additional tumor sections (PT13, PT14, and PT16; Supplementary Fig. 2d–f). All samples, except PT13, were collected post-chemotherapy. In these sections, we identified tumor, stromal, ductular reaction, and normal hepatocyte regions (Fig. 2f). Tumor regions in PT13, PT14, and PT16 exhibited fetal tumor signatures, characterized by broad expression of pericentral hepatic and fetal liver markers, and reduced expression of periportal markers (Fig. 2g, h and Supplementary Fig. 2g). Unlike post-chemotherapy tumors, which showed considerable necrotic regions and reduced heterogeneity, PT13 displayed more heterogeneity, likely due to its

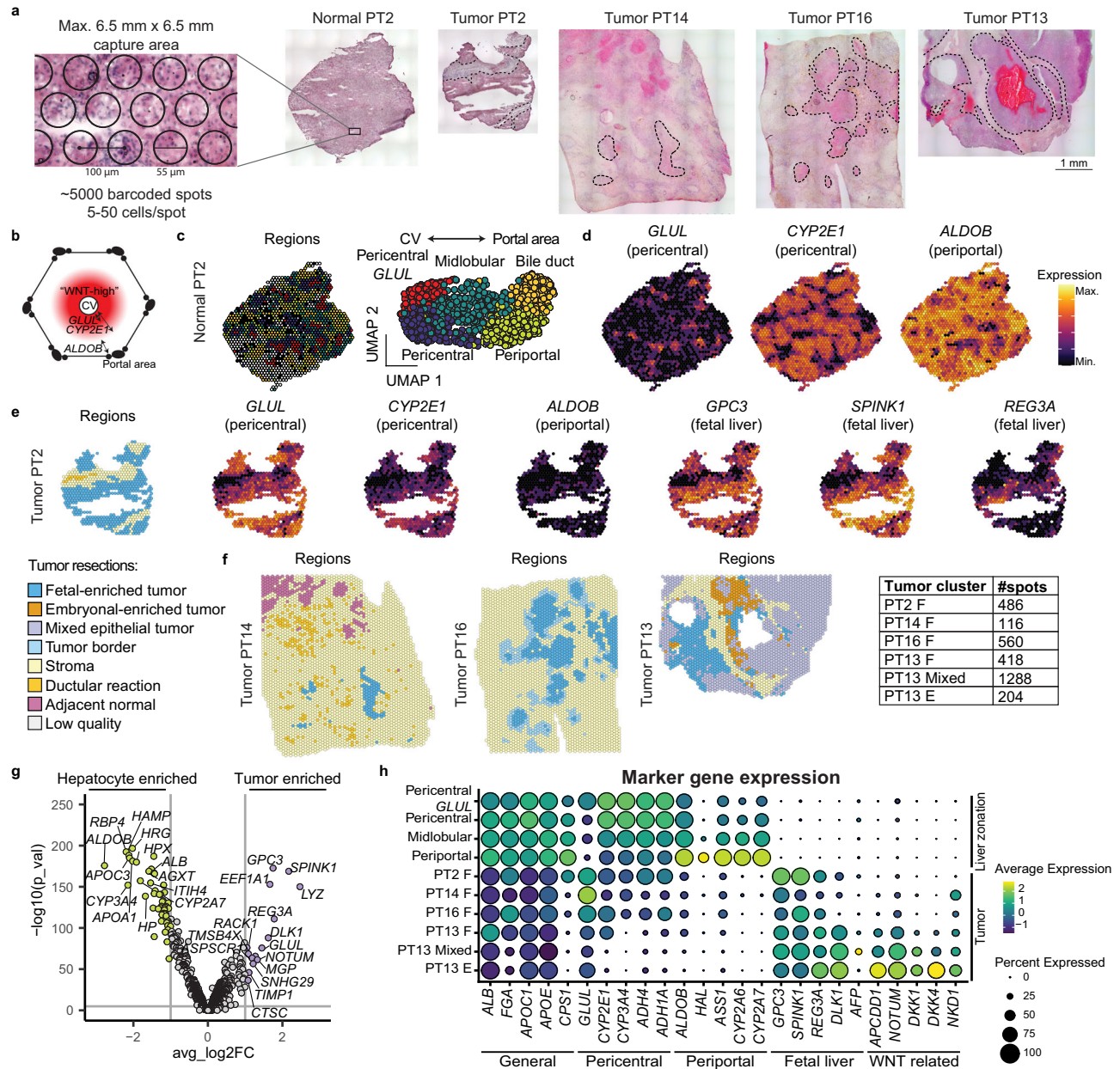

**Fig. 2 | Spatial transcriptomics of hepatoblastoma tissues displays distinct spatial molecular patterns. a** ST analysis was performed using the *10x Genomics Visium* platform on four hepatoblastoma tissues, as well as one paired distal normal liver tissue. H&E staining tissue sections are shown, with the tumor regions indicated (dotted line). **b** Schematic representation of a liver lobule showing a "WNT-high" gradient around the central vein (CV), associated with high *GLUL* (narrow) and *CYP2E1* (broad) expression, while the periportal zone is associated with a distinct hepatic expression profile, including expression of genes such as *ALDOB*. **c** Spatial map (left) and UMAP (right) annotated by liver metabolic zonation clusters. **d** Spatial map of liver zonation markers showing distinct expression patterns of pericentral and periportal markers in normal liver. **e** Spatial map of clusters and liver markers showing retained

(reduced) expression of pericentral markers, reduced expression of periportal markers, and expression of fetal liver markers in tumor regions of PT2, but absent in stromal regions. **f** Spatial map of clusters in three additional hepatoblastoma resections (PT14, PT16, PT13), showing fetal tumor regions. Additional heterogeneity is observed in PT13, where we identified fetal-enriched, embryonal-enriched, and mixed tumor regions. **g** Volcano plot of differentially expressed genes between all tumor regions and distal hepatocytes (pericentral, midlobular, and periportal) highlighting the expression of tumor-specific fetal liver markers and reduced expression of mature hepatic and periportal markers. *P*-values calculated using the Wilcoxon Rank Sum Test. **h** Dot plot comparing expression of hepatic markers between hepatocyte regions and tumor regions.

untreated nature. In addition to regions exhibiting fetal tumor characteristics, we also identified areas with embryonal tumor features in this tissue, characterized by high levels of WNT target genes such as *NOTUM, DKK1/4, NKD1, APCDD1* (Fig. 2h and Supplementary Fig. 2d). Furthermore, most of the tumor area expressed both fetal and embryonal tumor markers, indicating that the transcriptomic spots contained a mixture of both epithelial tumor cell types. These spots were annotated as mixed epithelial tumor regions. The expression profile of PT13 detected by ST correlates with scRNA-seq analysis and

revealed additional spatial heterogeneity (Fig. 1f and Supplementary Fig. 1g). Across the four tumor samples, inter-tumoral heterogeneity was also observed (Supplementary Fig. 2h). Overall, our ST analysis is consistent with fetal and embryonal tumor signatures.

### HNF4A and LEF1 mark distinct tumor subpopulations

To validate the presence of fetal and embryonal tumor clusters identified through our scRNA-seq and spatial analyses, we conducted immunofluorescence (IF) staining on a limited series of FFPE pre- and

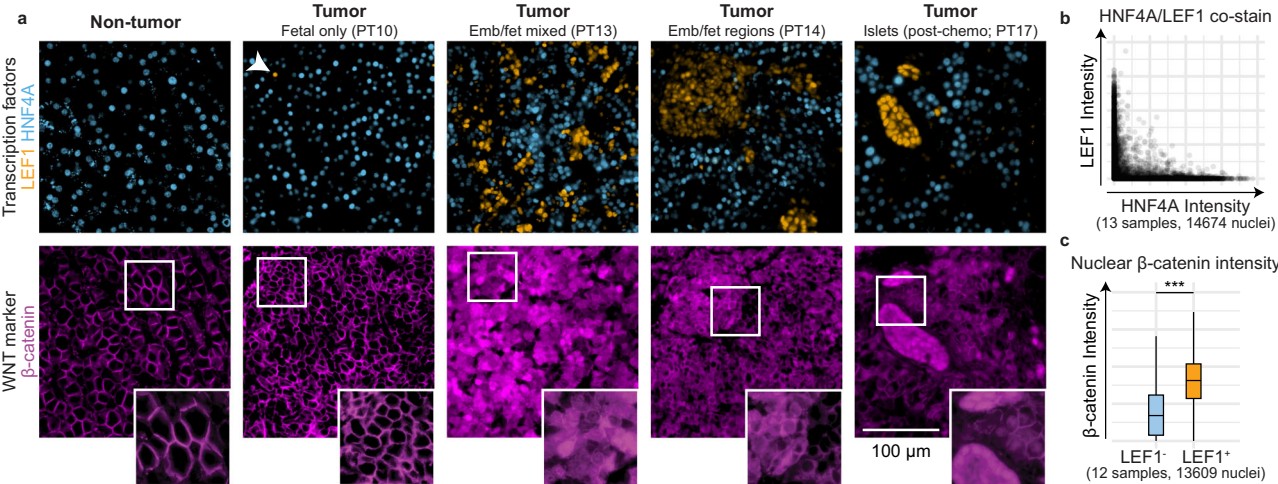

**Fig. 3 | Immunofluorescence staining of TFs LEF1 and HNF4A mark distinct tumor subpopulations. a** Co-staining of HNF4A (blue) and LEF1 (orange) in normal liver and tumor tissues displayed a mutually exclusive expression pattern, with different patterns shown (top). Arrowhead indicates a singular LEF1+ cell. Immunofluorescence staining of β-catenin on a consecutive section (bottom). Tissues from different patients ($n = 13$) and regions are available in Supp. Figure 3b. **b** Scatter plot showing HNF4A and LEF1 staining intensity per cell in hepatoblastoma tissues by nuclear segmentation and signal quantification. **c** Boxplots showing nuclear intensity of β-catenin staining in LEF1- and LEF+ cells by nuclear segmentation and signal quantification. Boxes represent the interquartile range, with the middle line showing the median. Whiskers extend to the smallest and largest values within 1.5 times this range. Welch's two-sample *t* test, ***$p < 0.0001$. Source data are provided as a Source Data file.

post-chemotherapy tumor samples collected from ten patients (Supplementary Table 1). Based on the results of our gene regulatory network analysis, we stained for the TFs HNF4A, the central hepatic regulator in the liver[41], and LEF1, a key mediator in the WNT/β-catenin pathway[35]. We were able to identify tumor cells that stained strongly for either HNF4A or LEF1, with the staining pattern of the two TFs being mutually exclusive (Fig. 3a, b and Supplementary Fig. 3a–c). We observed distinct regions of HNF4A+ cells and LEF1+ tumor cells, but also LEF1+ clusters that were interspersed with HNF4A+ clusters, as illustrated in PT13. Of note, islets of LEF1+ cells separated by stromal cells from surrounding HNF4A+ cells were also observed in post-chemotherapy samples (PT9, 17) (Fig. 3a and Supplementary Fig. 3b). The presence of HNF4A+ and LEF1+ tumor clusters was generally consistent with the mixed embryonal and fetal histological classification made by the pathologist (R.R.d.K., Supplementary Table 1). In most of these tumors, we observed both HNF4A+ and LEF1+ cells, while in samples annotated as 'predominantly fetal', we observed mainly HNF4A+ staining patterns, with only a few LEF1+ cells noted.

We further analyzed the β-catenin staining patterns in tumor tissues and one distal normal tissue. HNF4A+ regions showed a typical β-catenin membrane localization, as well as nuclear and cytoplasmic staining patterns not observed in the normal liver (Fig. 3a and Supplementary Fig. 3b). In contrast, LEF1+ regions generally displayed a more prominent nuclear and cytoplasmic β-catenin distribution, with less membrane staining compared to HNF4A+ regions, although there is substantial variation between patient samples. Based on quantitative analysis, on average, LEF1+ cells showed higher levels of β-catenin nuclear accumulation than LEF- cells (Fig. 3c). Overall, we confirmed the existence of two distinct tumor subpopulations expressing either HNF4A or LEF1 and found that these two markers could distinguish tumor subpopulations better than β-catenin staining alone.

## Establishment of patient-derived hepatoblastoma organoids across clinical stages

The limited availability of pediatric liver tumor models that recapitulate the genomic and transcriptomic heterogeneity of the disease has hindered in vitro modeling. Herein, we established hepatoblastoma tumor organoids (HBTOs) using patient tumor material,

obtained from biopsies or resected tumors. In total, we established organoid models from twelve patients, representing various clinical stages such as pre-chemotherapy, post-chemotherapy, relapse, and metastasis (Fig. 4a, Supplementary Fig. 4a, b, and Supp. Table 1). For two patients, PT13 and PT17, we established organoids from tumor material obtained at two different time-points: at diagnosis and during surgical resection.

To confirm the tumor origin of the organoids, we performed targeted sequencing on *CTNNB1* exon 3 and verified the presence of point mutations or deletions, as identified in the original tumors, in all organoids (Supplementary Table 1). Western blot analysis confirmed the reduction in β-catenin molecular weight in the organoids containing *CTNNB1* deletions (Supplementary Fig. 4c). In addition, inferred copy number variation (CNV) profiles illustrated the presence of chromosomal aberrations (Supplementary Fig. 4d).

## HBTOs recapitulate fetal and embryonal tumor features

To characterize the transcriptomic landscape of the organoid cohort, we performed scRNA-seq (Fig. 4). Unsupervised graph-based clustering showed that organoids primarily clustered according to their sample origins (Supplementary Fig. 4e, f). However, principal component analysis (PCA) segregated organoids from different patients into three groups. Based on marker expression, they were annotated as embryonal ('E') for the hepatic-low and WNT-high group, fetal-I ('F₁') for the hepatic-intermediate group, and fetal-II ('F₂') for the hepatic-high group (Fig. 4a, b).

We compared the expression patterns across the organoid cohort (Fig. 4c and Supplementary Fig. 5g). Hepatic TFs, hepatic functional markers (e.g., coagulation, fatty acid metabolism), and fetal liver markers were expressed at higher levels in fetal-II than fetal-I tumor organoids. These markers were low or absent in embryonal tumor organoids (Fig. 4c, d and Supplementary Fig. 4g). These markers correspond with the fetal tumor signature from tumor tissue analyses (Fig. 4e). The embryonal HBTOs were enriched with WNT target genes and EMT markers, including *AXIN2*, *LEF1*, *DKK1*, *NOTUM*, *NKD1* and *VIM*, in line with the embryonal tumor signature. In addition, fetal-I tumor organoids (hepatic-intermediate) expressed markers related to tumor progression and invasion, including

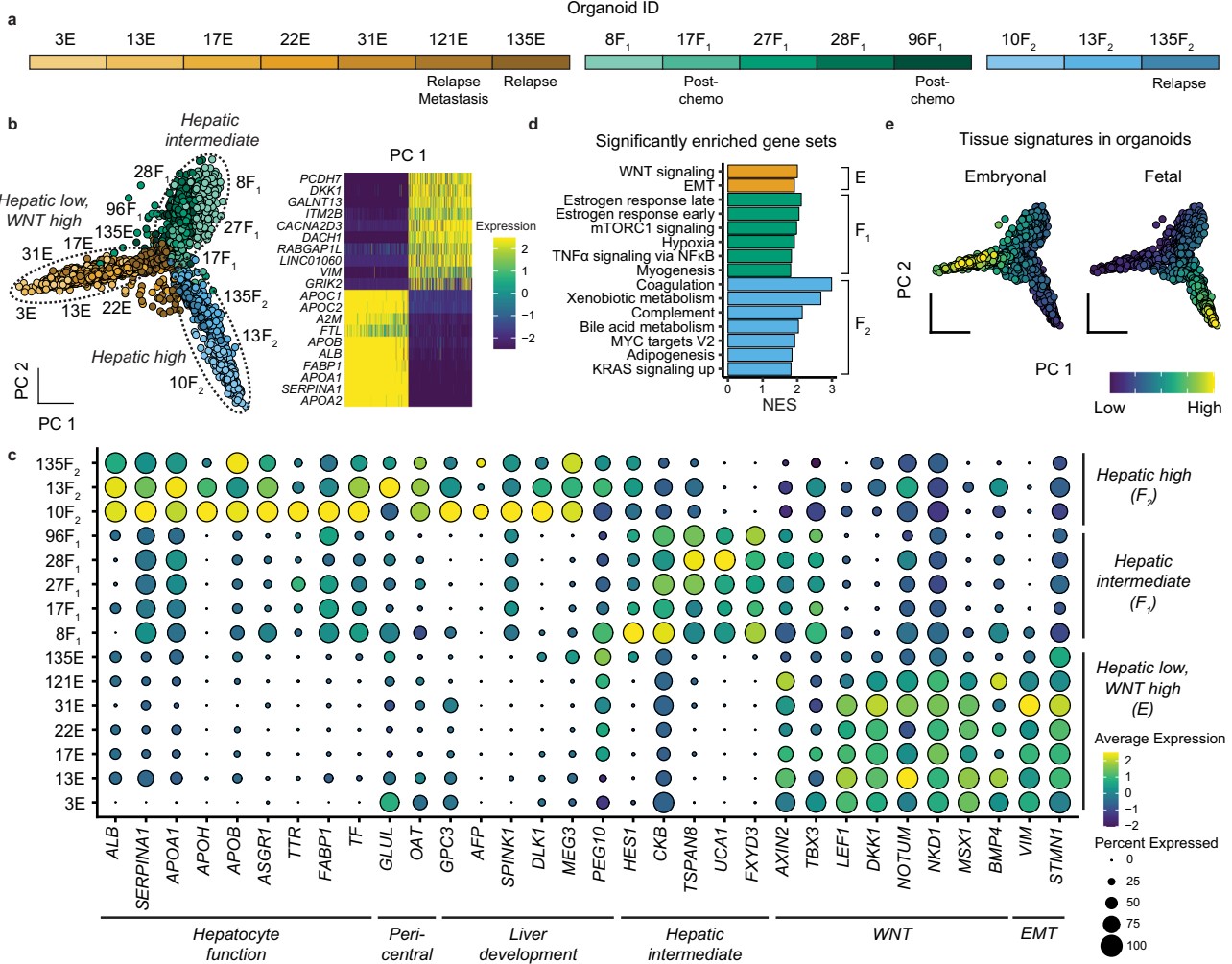

**Fig. 4 | Single-cell RNA-seq analysis elucidates molecular heterogeneity of hepatoblastoma tumor organoids. a** Overview of different organoid lines and their corresponding clinical stages. **b** PCA plot illustrating the clustering of three organoid groups based on hepatic marker expression levels (left); heatmap showing top markers in the first principal component (right). **c** Dot plot showing the expression of markers for the different groups of organoids. **d** Gene set enrichment analysis of hallmark gene sets of the three organoid groups, calculated using the fgsea package, with an adjusted *p*-value < 0.05. **e** PCA plot showing fetal and embryonal signatures derived from the tissue scRNA-seq analysis in the organoids.

*TSPAN8*, *CKB*, *UCA1,* and *FXYD3* (Fig. 4c). Of relevant note, two of these samples (17F$_1$ and 96F$_1$) were derived from post-chemotherapy tumor material. These organoids, except 8F$_1$, could only be cultured in a different medium ('reduced medium', see "Methods" and Supplementary Table 1). Further studies are needed to investigate the biological implications of these observations.

We performed gene regulatory network analysis on this dataset and identified enrichment of WNT pathway regulons, specifically LEF1 and TCF7, in the embryonal tumor organoids, while HNF4A was enriched in the fetal-I and II tumor organoids (Fig. 5a and Supplementary Fig. 5a). Another hepatic regulon, FXR, essential for bile acid, lipid, and glucose metabolism, was enriched in the fetal-II organoids but absent in the fetal-I organoids (Fig. 5a), which may explain the reduced hepatic features in the fetal-I group. Moreover, unbiased clustering based on regulon activity segregated organoids into three main groups rather than according to patient origin (Fig. 5b). In summary, our analysis of the organoid cohort revealed that the expression patterns of hepatic markers, WNT pathway-related factors, and EMT markers aligned with the fetal and embryonal hepatoblastoma signatures identified in the tumor tissue analyses. This suggests that our organoid models effectively recapitulate the characteristics of the distinct subpopulations observed in hepatoblastomas.

Next, to validate the gene regulatory network analysis, we performed single-cell Multiome ATAC and gene expression analysis on a subset of HBTOs (Fig. 5c, d). The differential gene expression per sample is shown in Supplementary Fig. 5b. ScATAC analysis identified enrichment of TF motifs for hepatic development in the fetal-I and II tumor organoids, such as HNF family members (*HNF4A*, *4 G*, *1B*, *1 A*), *RXRG* and *PPARD* (Fig. 5d). Furthermore, the fetal-I (hepatic-intermediate) tumor organoids showed enrichment of motifs of AP-1 family TFs (*FOS/JUN*), which are typically activated during liver regeneration[42]. The embryonal tumor organoids were enriched in TF motifs related to the WNT pathway (*LEF1*, *TCF7*, *TCF7L2*), EMT (*MEOX2*), and apoptosis (*TP53*, *TP63*, *TP73*).

We further validated the expression of HNF4A and LEF1 in HBTOs cultured over various passages (P2-10) by IF staining (Fig. 5e and Supplementary Fig. 5c). In the fetal-I and II tumor organoids, HNF4A expression was consistently observed throughout, while LEF1 expression was generally absent. Conversely, the embryonal tumor organoids showed LEF1 expression broadly, with HNF4A largely absent. We also performed scRNA-seq on one organoid model (13F$_2$) at an early (P8) and at a later passage (P20), and found they still clustered together (Supplementary Fig. 4f). These findings suggest stable subtype-specific expression profiles in HBTOs during extended periods of culture.

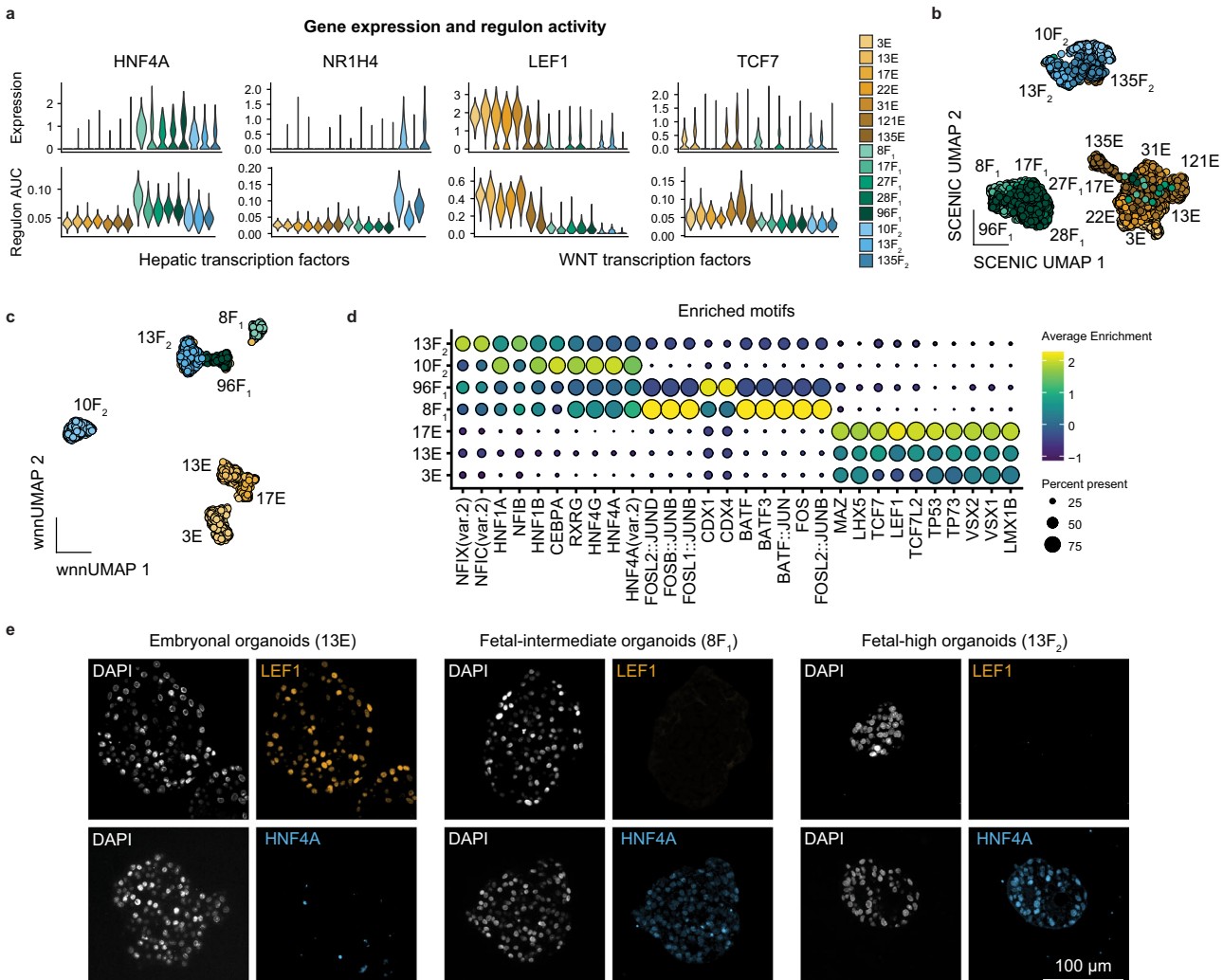

**Fig. 5 | Single-cell ATAC-seq and SCENIC analysis identify TFs associated with hepatoblastoma subtypes. a** Violin plots showing expression levels and SCENIC regulon activity scores for hepatic transcription factors (HNF4A, NR1H4) and WNT/β-catenin transcription factors (LEF1, TCF7). **b** UMAP visualization based on SCENIC regulon activity scores of the different organoids. **c** Weighted nearest neighbor (WNN) UMAP visualization based on the integration of RNA and ATAC-seq data, showing clustering per organoid model. **d** Dot plot showing top 5 enriched motifs per organoid model from ATAC-seq analysis. **e** Immunofluorescence staining of FFPE-sectioned slides for organoids for LEF1 and HNF4A, with representative images from the three organoid groups.

In addition, we performed IF staining for β-catenin and confirmed its localization in the membrane, nucleus, and cytoplasm in tumor organoids (Supplementary Fig. 5d).

In sum, we were able to expand tumor organoids from the two major hepatoblastoma subtypes. The transcriptomic and chromatin accessibility landscapes of the organoids correspond with the fetal and embryonal tumor signatures (Figs. 1–3). In addition, consistent with tumor tissue stainings, HNF4A and LEF1 marked fetal and embryonal tumor organoids, respectively (Fig. 5e and Supplementary Fig. 5c).

### Drug screening identifies HDACs as potential therapeutic targets

To identify potential targeted treatment options against hepatoblastoma, we performed drug screening of over 200 compounds on eleven of our tumor organoid models (Fig. 6, Supplementary Table 1 and Supplementary Data 2). The use of this library is supported by previous studies conducted at our center[43–47]. To assess drug sensitivity, the area under the curve (AUC) for the dose-response curve of each compound was determined. These findings were depicted in a hierarchically clustered heatmap, providing insights into the drug

sensitivity profiles and highlighting correlations between the different organoid groups (Fig. 6a and Supplementary Fig. 6a).

We identified various classes of inhibitors targeting a wide range of hepatoblastoma organoids, including those that target HDAC, the proteasome, PLK-1, and FGFRs (Fig. 6b). Some of these compounds, such as vorinostat[48] (an HDAC inhibitor), bortezomib[19] (a proteasome inhibitor), and volasertib[49] (a PLK-1 kinase inhibitor), were previously found to be effective against liver tumor cell lines and PDX models (Fig. 6a, b). However, compounds identified from other studies, such as olaparib[50], a PARP1 inhibitor, were only moderately effective in our organoid models, indicating potential biological differences between tumor cell lines and patient-derived organoid models (Supplementary Fig. 6b).

We further investigated the effect of different HDAC inhibitors on the organoid models. The organoids were sensitive to romidepsin (targeting HDAC1/2), panobinostat (pan-HDAC), and fimepinostat (HDAC1/2/3/10), but not to entinostat (HDAC1/3) or PCI-34051 (HDAC8) treatment (Fig. 6b–d, Supplementary Fig. 6b and Supplementary Table 2). This finding indicates that only a subset of HDAC inhibitors may have therapeutic utility in hepatoblastoma, in agreement with a previous study[48]. The HDAC protein family consists of

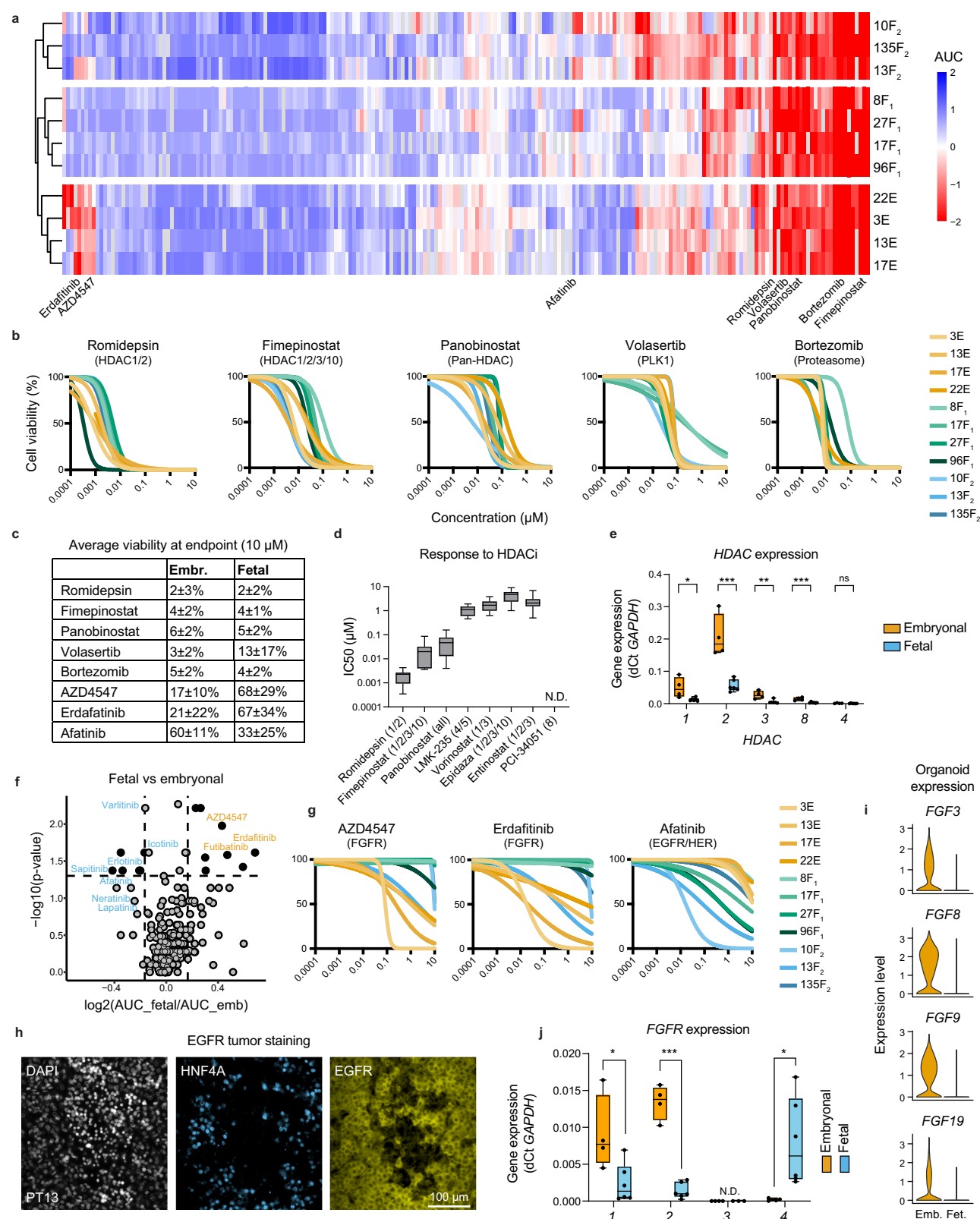

eleven HDACs, divided into four subclasses based on sequence homologies. We assessed the mRNA expression of HDACs through scRNA-seq analysis and observed the expression of *HDAC1,2,* and, to a lesser extent, *HDAC3* (Supplementary Fig. 6c).

We further assessed the mRNA expression of all Class I HDACs (*HDAC1, 2, 3* and *8*), as well as previously reported *HDAC4* (a Class IIa HDAC) in tumor organoids using qRT-PCR (Fig. 6e).

*HDAC1* and *HDAC2* showed high expression levels across the cohort, with *HDAC2* expression being approximately 10-fold higher than that of *HDAC1*. The expression of *HDAC1/2* was also higher in embryonal tumor organoids than fetal tumor organoids (Fig. 6e and Supplementary Fig. 6c). *HDAC2* expression was also described previously by Cairo et al.[14] Other HDACs, including *HDAC3, 8,* and *4* were expressed at lower levels than *HDAC1* and *2*.

**Fig. 6 | High-throughput drug screening of tumor organoids provides insight into targetable pathways. a** Clustered heatmap, scaled by rows, of AUC values from the dose-response curves in high-throughput drug screens across 11 organoid models. **b** Dose-response curves of selected drugs effective in all models, such as those targeting HDAC, PLK1, and proteasomes. **c** Table showing the average viability (%) at the highest concentration (10 μM) for both fetal and embryonal tumor organoids. **d** Boxplot showing IC$_{50}$ values for all organoid models ($n = 11$) against HDAC inhibitors included in the drug library. The boxes represent the interquartile range, the line inside the box marks the median, and the whiskers extend to the lowest and highest values. **e** Boxplot showing the expression levels of HDACs across embryonal ($n = 4$) and fetal ($n = 6$) organoid models measured by qRT-PCR. The boxes represent the interquartile range, the line inside the box marks the median, and the whiskers extend to the lowest and highest values. **f** Volcano plot identifying the most effective compounds specific for either the embryonal (orange) or fetal (blue) tumor organoid

models. *P*-values were calculated using the Wilcoxon Rank Sum Test. **g** Dose-response curves of selected drugs that show selective sensitivities in embryonal lines (targeting FGFR) and fetal lines (targeting EGFR/HER). **h** Immunofluorescence co-staining of HNF4A and EGFR in a representative tumor sample. **i** Expression of select FGF ligands in embryonal and fetal organoids, as measured by scRNA-seq. **j** Boxplot showing the distribution of FGFRs across embryonal and fetal models in organoids measured by qRT-PCR. The boxes represent the interquartile range, the line inside the box marks the median, and the whiskers extend to the lowest and highest values. Statistical significance for (**e**) and (**j**) was determined using an unpaired, two-sided *t* test. *$p < 0.05$; **$p < 0.01$; ***$p < 0.001$; ns not significant; N.D. not determined. Exact *P*-values: *HDAC1* = 0.02, *HDAC2* = 0.0006, *HDAC3* = 0.006, *HDAC8* = 0.001, *HDAC4* = 0.199, *FGFR1* = 0.025, *FGFR2* < 0.0001, *FGFR3* = 0.071, *FGFR4* = 0.034. Source data are provided as a Source Data file.

These data suggest that HDAC1 and 2 could be the possible therapeutic target of HDAC inhibitors in hepatoblastoma.

### Distinct drug sensitivity profiles between fetal and embryonal tumor organoids

We compared drug response profiles of fetal and embryonal tumor organoids and identified distinct sensitivities to targeted inhibitors (Fig. 6f, g). In particular, fetal tumor organoids were sensitive to the treatment of small molecule inhibitors targeting the EGFR/HER, such as afatinib, erlotinib, and sapitinib, while embryonal tumor organoids were sensitive to pan-FGFR inhibitors, such as erdafitinib, futibatinib, and ponatinib (Fig. 6f, g, Supplementary Fig. 6b and Supplementary Table 2).

The EGF and FGF signaling pathways are critical for cell survival and proliferation[51,52]. Therefore, our culture medium included growth factors such as EGF, FGF10, and HGF. Given the sensitivity of fetal tumor organoids to EGFR inhibitors, we asked whether EGF is essential for organoid culture. EGF withdrawal significantly reduced the growth of fetal HBTOs by approximately 50% but did not affect the growth of embryonal tumor organoids (Supplementary Fig. 6d). The expression of EGFR was observed in the fetal component of tumor tissues by scRNA-seq (Fig. 1c). We further validated the expression of EGFR by IF staining, showing EGFR staining primarily in HNF4A$^+$ regions, underscoring the critical role of EGF signaling in fetal tumor cells (Fig. 6h and Supplementary Fig. 6e).

In addition, the removal of FGF10 from the culture medium did not reduce the growth of tumor organoids, suggesting that FGF signaling in these organoids might be predominantly autocrine (Supplementary Fig. 6d and Supplementary Methods). This is supported by scRNA-seq data showing the expression of *FGFR1* and various *FGF* ligands (e.g., *FGF3, 8, 9, 19*) in tumor tissues and organoids (Figs. 1c, 6i and Supplementary Fig. 6f). We assessed the expression of *FGFR1–4* in organoids by qRT-PCR. We detected expression of *FGFR1* and *FGFR2* in tumor organoids, with higher levels in embryonal than fetal organoids, while *FGFR4* was exclusively expressed in fetal tumor organoids (Fig. 6j and Supplementary Fig. 6f).

Next, we assessed the sensitivity of tumor organoids to FGFR inhibitor treatment when cultured with and without FGF10. We found that the removal of FGF10 did not alter the response of either fetal or embryonal organoids to the FGFR inhibitor erdafitinib (Supplementary Fig. 6g and Supplementary Methods). Embryonal organoids were susceptible to FGFR inhibitors at various inhibitor concentrations. By contrast, fetal organoids only displayed sensitivity to FGFR inhibitors at the highest concentration of the inhibitor. Although *FGFR4* was expressed in fetal tumor organoids, FGF19 was not present in our culture medium, indicating that it was not essential for maintaining organoid culture in the presence of other growth factors. Overall, these findings suggest a previously unrecognized dependency on EGF and FGF signaling pathways in the tumorigenesis of hepatoblastoma.

### Drug sensitivity profiles of HB compared with other pediatric tumors

To identify compounds that specifically target hepatoblastoma, we compared the drug sensitivity profiles of hepatoblastoma organoids to a reference cohort consisting of other pediatric tumor models, which included Wilms tumor[43], malignant rhabdoid tumor[44], rhabdomyosarcoma[47], neuroblastoma[45], and Ewing sarcoma models[46] (Supplementary Fig. 6h and Supplementary Table 3). We found that romidepsin was highly specific for hepatoblastoma compared to the reference cohort. Furthermore, we found that FGFR and EGFR inhibitors were specific for the respective hepatoblastoma subtypes compared to the reference cohort. In summary, our study identified drugs that could target both fetal and embryonal tumor organoids, as well as drugs that could selectively target a specific hepatoblastoma subtype.

## Discussion

The majority of hepatoblastoma cases (> 90%) carry a mutation in the *CTNNB1* gene encoding for β-catenin. However, hepatoblastoma still presents significant molecular heterogeneity[2,53]. In this study, we employed scRNA-seq/ATAC-seq, ST, and gene regulatory network analysis to examine patient tumor tissues and tumor organoids. We described two distinct tumor signatures corresponding to fetal and embryonal tumor subtypes (Fig. 7).

The 'hepatic' fetal tumor cells expressed hepatic markers related to liver function, including metabolic markers activated by the WNT pathway[23] and previously described fetal liver markers[14]. Gene regulatory network analysis showed that hepatic TFs, such as ones regulating liver development and function (HNF4A), bile acid and lipid metabolism (FXR, AR), drug metabolism, and detoxification (CAR, PXR), are predominantly enriched in the fetal group. Of note, in the normal liver, WNT/R-spondin signals originating from endothelial cells of the central vein regulate the expression of these pericentral markers in hepatocytes, including drug and ammonia metabolism[54–56]. In this context, fetal tumor cells exhibit characteristics of pericentral hepatocytes.

Compared to fetal tumor cells, the 'WNT-high' embryonal tumor cells displayed low levels of hepatic markers but showed high levels of general WNT pathway target genes (*AXIN2, APCDD1, NOTUM*, and *NKD1*), EMT markers (*VIM*), the p53 pathway, and mitotic spindle-related genes. This group was enriched in key WNT pathway TFs, such as LEF1, TCF7, and TCF7L2, which was corroborated by the chromatin accessibility landscape analysis of a cohort of tumor organoids established in this study.

The fetal tumor signature correlates with hepatic tumor signatures previously identified as 'C1' by Cairo et al.[14], 'hepatocytic' by Hirsch et al.[17], and 'hepatocyte' by Nagae et al.[18]. Compared to fetal tumor cells, the embryonal tumor signature correlates with previously described 'progenitor' tumor signatures identified as 'C2' by Cairo et al., 'liver progenitor' by Hirsch et al. or 'proliferative' by Nagae et al. Cairo et al. noted the enrichment of general WNT

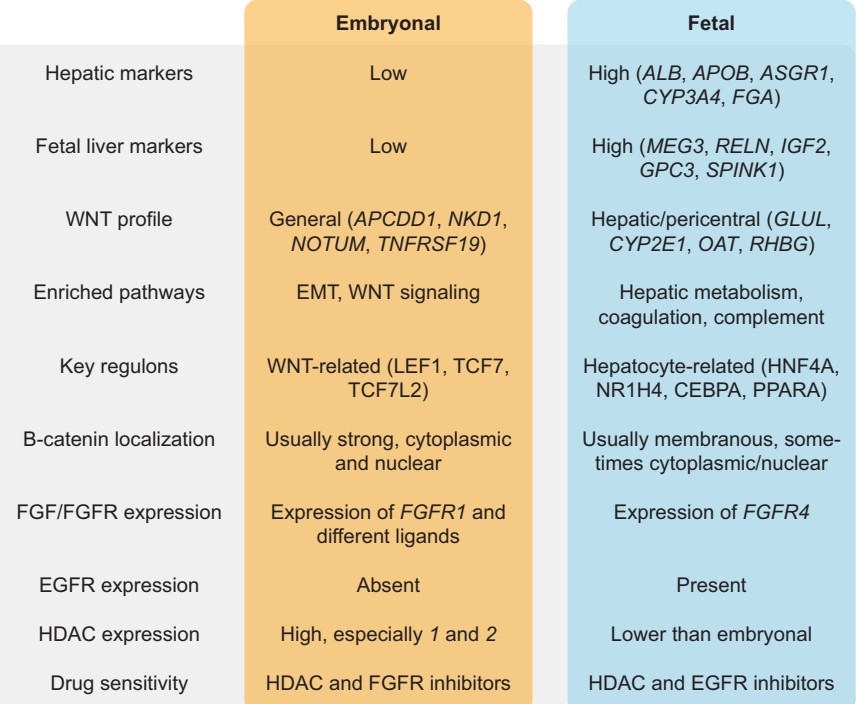

**Fig. 7 | Summary of the characteristics of embryonal and fetal tumor components in hepatoblastoma.**

signaling markers broadly across tumor samples when compared to normal liver, and Nagae et al. described the enrichment of general WNT markers in their 'proliferative' subtype of tumors, which likely correspond with embryonal-enriched tumors. However, minimal overlap was observed between tumor signatures described in this study and those from previous studies. Specifically, we described a strong WNT target gene enrichment in the embryonal cells. This finding is more consistent with a recent study by Wu et al.[57], which profiled tumor tissue sections with fetal or embryonal histology obtained by laser capture microdissection.

Furthermore, we demonstrated a mutually exclusive expression pattern of HNF4A and LEF1 in a cohort of tumor tissues by IF staining. Our data suggests dichotomous expression patterns in hepatoblastoma, with intra- and inter-tumoral heterogeneity within each tumor subset. The absence of HNF4A in the embryonal tumor subset is intriguing because HNF4A is generally considered a central regulator of hepatic differentiation and essential for maintaining liver function[41,58]. In previous studies, the knockdown of *HNF4A* during hepatic development prohibited the expression of many hepatic genes[59]. The absence of HNF4A and other hepatic TFs may explain the low levels of hepatic markers in the embryonal tumors. Moreover, HNF4A has been shown to be essential for the transition of endoderm to a hepatic fate during development[59,60]. These observations suggest that hepatoblastoma might arise from an early fetal liver developmental stage, with cellular differentiation being halted along the developmental trajectory.

In the normal liver, hepatocytes displayed predominantly membranous β-catenin staining. However, mutations in the *CTNNB1* gene within tumor cells result in the intracellular accumulation of β-catenin molecules. Previous studies[14,61] observed that embryonal regions correlate with diffuse nuclear and cytoplasmic β-catenin staining, while fetal regions correlate with moderate cytoplasmic and membranous staining along with focal nuclear staining. This is generally in agreement with our own observations, although relying on the β-catenin staining pattern alone for distinguishing tumor

subtypes is challenging. In addition, we observed a stronger nuclear β-catenin staining in LEF1+ cells compared to LEF1− cells. In summary, based on the staining pattern for HNF4A and LEF1, we were able to unequivocally distinguish these two hepatoblastoma subtypes, while it was not evident with β-catenin staining alone.

The histological classification of hepatoblastoma is often complicated due to tumor heterogeneity. Furthermore, IF staining of HNF4A/LEF1 revealed considerable spatial heterogeneity. For instance, in some tumor tissues, LEF1+ and HNF4A+ clusters were diffusely intermixed, whereas they were more separated in others. Moreover, isolated islets of LEF1+ cells were observed predominantly in post-chemo tumor tissues. We propose that incorporating LEF1 and HNF4A into diagnostic staining panels could enhance tumor diagnosis, enabling a more precise tumor subtyping and correlations between molecular profiles of tumor and disease prognosis.

The distinct WNT signaling programs observed in the fetal and embryonal tumor cells are noteworthy. A previous study focusing on liver zonation described the interaction between HNF4A, β-catenin, and TCF7L2. Briefly, HNF4A can repress β-catenin-dependent transcription, while β-catenin can repress HNF4A-dependent transcription[34]. Another study using hepatocellular carcinoma (HCC) cells found that HNF4A competes with β-catenin for binding to TCF7L2 to facilitate repression of β-catenin/TCF7L2 target genes[62]. In addition, overexpression of HNF4A resulted in the depletion of nuclear β-catenin. Although the relationship between β-catenin and HNF4A in hepatoblastoma remains to be elucidated, a similar mechanism could be at play here.

In addition, a recent study by Pagella et al.[38] demonstrated that β-catenin binding to genomic loci is cell-type specific and associated with both activation and repression of gene expression programs, leading to divergent responses depending on the cellular context, and that β-catenin DNA binding is a highly dynamic and temporally regulated process.

We established a large cohort of hepatoblastoma organoids across clinical stages (i.e., pre-, post-chemotherapy, relapse), which facilitated drug screening using an extensive panel of compounds and

illustrates the presence of distinct drug sensitivity profiles correlating with tumor heterogeneity. These profiles correlate with tumor subtypes rather than the clinical stage. For instance, fetal tumor organoids were selectively targeted by EGFR inhibitors, whereas embryonal organoids were targeted by FGFR inhibitors.

EGF and FGF signaling are potent mitotic growth factors essential for cell proliferation during liver regeneration and for in vitro hepatocyte culture[51,52,63,64]. We observed that EGFR expression was restricted to the fetal tumor regions. EGF is normally produced in the Brunner's gland in the duodenum and reaches the liver via the portal circulation[63]. By contrast, the expression of *FGFR1* and several *FGF* ligands were observed in embryonal tumor cells and organoids, notably *FGF8*, which was previously identified as one of the most upregulated genes in metastatic hepatoblastoma[65]. In addition, *FGFR1* was previously found to be upregulated in hepatocytes by WNT signaling, based on the *APC^{-/-}* mouse model[23]. An intriguing observation is the expression of *FGFR4* in the fetal tumor cells and its ligand *FGF19* in the embryonal tumor cells, suggesting crosstalk between the two subtypes. Indeed, this interaction was described in the study by Wu et al.[57] Taken together, the transcriptomic profiles and drug screening results suggest the involvement of EGF/FGF pathways in hepatoblastoma tumorigenesis.

We further identified multiple compounds from various classes of inhibitors that could target a broad range of organoids, such as HDAC inhibitors, proteasome inhibitors, and PLK-1 inhibitors. Of particular interest are HDAC inhibitors, most notably romidepsin, which targets HDAC1 and 2, and panobinostat, a pan-HDAC inhibitor. HDAC inhibitors have previously been proposed as a novel therapy against hepatoblastoma, and HDACs have been shown to be overexpressed in hepatoblastomas[48]. We confirmed *HDAC1/2* expression in tumor organoids, with the embryonal tumors expressing higher levels of HDACs than the fetal tumors. Previously, HDAC1 has been shown to interact with LEF1 to regulate its activity[66]. In the same study by Pagella et al.[38], β-catenin was found to be dependent on HDAC for chromatin opening, and HDAC inhibition fully abolished its activity. Furthermore, *HDAC1/2* is crucial for liver regeneration, and loss of these factors impaired hepatocyte proliferation in a mouse study. Finally, the compounds identified in this study, including romidepsin and panobinostat, have been approved for clinical use[67–70], presenting a great opportunity to assess their efficacy in pediatric liver cancer patients.

## Methods

This research projects complies with all relevant ethical regulations. Ethics approval was granted for the biobanking initiative by the medical ethics committee of the Erasmus Medical Center (Rotterdam, the Netherlands; MEC-2016-739) and the Universitätsklinikum Münster (2017-261-f-S). This project was approved by the Princess Máxima Center Biobank and Data Access Committee (PMCLAB2020-107).

### Primary tumor tissues

Fresh tumor material and distal normal liver tissue were obtained with written informed consent from biopsies and resections performed at the Princess Máxima Center for Pediatric Oncology Utrecht and the University Medical Center Groningen. Patient samples and clinical data, including age and sex, were obtained after approval by the Biobank and Data Access Committee. For all tissues, H&E and standard IHC stainings for hepatoblastoma were performed in the diagnostic context and reviewed by a pathologist (R.R.d.K.). Viably frozen tumor samples were obtained from the Universitätsklinikum Münster for organoid establishment.

### Organoid culture

Tissue dissociation was performed as described previously[21]. Briefly, tissues were minced into 1-2 mm pieces using small scissors in a glass Petri dish, followed by incubation in pre-warmed Liver Perfusion Medium (Thermo Fisher) for 15 min at 37 °C. Next, minced tissues were washed once with DPBS and spun down at $300 \times g$ for 3 min. The pellet was collected and incubated in a pre-warmed digestion mix at 37 °C on a shaker at 150 rpm. The digestion mix consisted of Liver Digestion Medium (Thermo Fisher) with 1% HEPES and 700 U/mL Collagenase type IV (Worthington Biochemical). Dissociation progress was assessed after 15 and 30 min, and the samples were mechanically dissociated by pipetting up and down 10 times to obtain substantial dissociation. If necessary, red blood cell lysis buffer (BioLegend) was used, following the manufacturer's instructions. Finally, the enzyme was washed off using Hepatocyte Wash Medium (Thermo Fisher), and the remaining cells or small aggregates were mixed with 100% reduced growth factor basement membrane extract (BME; R&D) and plated in 20 µL droplets on culture plates that had been incubated overnight. Occasionally, no digestion was necessary, and aggregates were plated immediately after mincing. Organoids were cultured in the culture medium described previously, with slight modifications[64,71,72]. The final culture medium (termed 'full medium') consisted of: Advanced DMEM/F12 (Gibco) containing HEPES (Gibco), GlutaMAX (Gibco), Penicillin-Streptomycin (Gibco), and Normocin (InvivoGen), supplemented with 2% B27 (Gibco), 1% N2 (Gibco), 1.25 mM N-Acetylcysteine (Sigma-Aldrich), 10 nM gastrin (Sigma-Aldrich), 50 ng/mL EGF (Peprotech), 10% RSPO1 conditioned media (produced in house), 100 ng/mL FGF10 (Peprotech), 25 ng/mL HGF (Peprotech), 10 mM Nicotinamide (Sigma-Aldrich), 5 µM A83.01 (Tocris), 10 µM forskolin (Tocris) and 10 µM Y27632 (Sigma-Aldrich) and 0.5 nM next-generation WNT surrogate[73]. In later established models, FGF10 was omitted and did not affect organoid establishment or growth (for the drug screens, this was only included in 10F_2, 13E, 13F_2, 17E, and 17F_1). Several lines (17F_1, 27F_1, 28F_1, 96F_1) could not be expanded in the 'full medium' without forming cystic organoids typically observed in cholangiocyte organoid culture. For these lines, we used the medium as described in Wu et al.[72] Compared to the 'full medium', the reduced medium lacks gastrin, forskolin, RSPO1 and WNT, and is supplemented with 3 nM dexamethasone (Bio-Techne). Organoids could be passaged every 1–3 weeks and varied between tumors. Passaging was performed as described before[74]. Briefly, BME was first dissociated using dispase (STEMCELL Technologies), before washing at least twice with PBS supplemented with 5% FBS (v/v). Organoids were then incubated in TryPLE Express (Gibco) at 37 °C pipetted to obtain small aggregates of cells and washed once more before replating in BME. See Supplementary Table 4 for passage numbers used for different experiments.

### Single-cell RNA and ATAC sequencing

Single-cell analysis of tissue and organoid samples was performed using the 10x Genomics Single-Cell Expression platforms according to the manufacturer's protocols (Chromium Next GEM Single-Cell 3' Reagent Kits v3.1 and Chromium Next GEM Single-Cell Multiome ATAC + Gene Expression). In brief, fresh tissues were minced (< 2 mm²) and viably frozen until processed. For scRNA-seq, minced tissue was rapidly defrosted, washed, and dissociated for 1 h at 37 °C, 250 rpm using dissociation mix 1. Dissociation mix 1 contained 0.5 mg/mL Liberase (Thermo Fisher) and 1 mg/mL Collagenase type IV (Gibco) in DMEM/F12 supplemented with Glutamax and 20 U/mL DNase (Thermo Fisher). Red blood cells and dead cells were removed using an RBC lysis buffer (BioLegend) and a Dead Cell Removal kit (Miltenyi Biotec), respectively. Prior to loading, the cells were passed through Flowmi cell strainers (40 µm, Merck) and counted using a Bürker chamber.

For the scRNA-seq of the organoids, 3E and 8F_1 organoids were sequenced individually, while the other lines were pooled together using the 3' CellPlex Multiplexing Kit. In brief, organoids were dissociated using TryPLE Express into single cells and incubated with a unique 10x CellPlex molecular tag for 15 min at RT while shaking (250 rpm) before washing and pooling. Demultiplexing was performed

during data analysis using Cell Ranger (v7.0.1), and all lines were integrated and analyzed together. For the organoid Multiome samples, organoids were first dissociated into single cells using TryPLE Express. Nuclei were isolated with NP40 lysis buffer with a 5 min incubation on ice and passed through a 70 µm cell strainer (Greiner Bio-One EASY-strainer, Thermo Fisher). Samples were pooled based on nuclei concentration (Countess II cell counter, Thermo Fisher) and sorted on a Sony SH800S cell sorter with a 100 µm nozzle for 7AAD (Invitrogen) positive singlets. After nuclei permeabilization, samples were counted (Countess II). Raw data was processed using Cell Ranger ARC (v2.0.0). SNP-based demultiplexing was performed using Python packages cellsnp lite (v1.2.2) and Vireo (v0.2.3). Genotyping references of the donors were obtained from whole-genome sequencing or whole-exome sequencing data of the tumor biopsy samples generated in the diagnostic setting. We recovered, on average, approximately 2200 and 1200 cells per organoid line for individually sequenced and multiplexed samples, respectively.

### Spatial RNA sequencing

Fresh tissues were snap-frozen in isopentane (Sigma-Aldrich) chilled by liquid nitrogen. Frozen tissue pieces were embedded in Tissue-Tek O.C.T. Compound (Sakura) and stored at −80 °C until cryo-sectioning. Tissues were selected based on tissue histology and RNA quality (RIN scores > 2.4 of TRIzol isolated RNA). ST was performed using the Visium Spatial Gene Expression Solution (10x Genomics) according to the manufacturer's protocols. In brief, 10 µm thick tissue sections were cut in a cryostat (Cryostar NX70, Thermo Fisher) and placed within the capture area of Visium Spatial Gene Expression Slides. Tissues were fixed in chilled methanol, and H&E stainings were performed to assess tissue morphology and quality. The slides were imaged using a brightfield microscope (Leica DMi8 S platform). Tissue permeabilization times for normal liver and tumor tissue were optimized using the Visium Spatial Tissue Optimization workflow. Permeabilization times were set at 12 min for tumor tissue and 18 min for normal liver tissue. Full-length cDNA and libraries were analyzed using a Qubit 4 fluorometer and an Agilent 2100 Bioanalyzer. cDNA libraries were sequenced on a NovaSeq6000 System (Illumina) with sequencing settings recommended by 10x Genomics.

### Single-cell RNA data analysis

Single-cell transcriptome data of hepatoblastoma and paired normal liver were obtained from Song et al.[21] and accessed through the Gene Expression Omnibus (GEO) database (accession number GSE186975). Raw UMI-collapsed read-count data was analyzed through Seurat[75] (v4.3.0) in R (v4.1.2). First, contamination by ambient RNA was estimated and removed by DecontX[76] (celda v1.10.0) using default settings. Next, low-quality cells were filtered by removing cells with fewer than 500 genes or a percentage of mitochondrial genes greater than 20%. In addition, a maximum threshold for the number of genes was set for individual samples. Separate workflows were implemented for dimensionality reduction and gene expression analyses. For dimensionality reduction and clustering, normalization was performed by SCTransform (v0.3.5), and cell-cycle and related genes were identified as previously described[43]. We then integrated the data from different batches using fastMNN batch correction (SeuratWrappers v0.3.1), while removing the cell cycle and related genes from the integration features. Dimensionality reduction and clustering were performed on the fastMNN-corrected data, using 40 dimensions and a resolution of 0.5, yielding 22 clusters. For gene expression analyses, counts were normalized and log-transformed by Seurat's "LogNormalize" method, followed by identification of the 2000 most variably expressed genes and linear transformation. Differentially expressed genes for each cluster were identified by Seurat's FindAllMarkers and three genes with the highest fold-change expression were selected for each cluster to facilitate cell type identification. All clusters containing tumor cells or epithelial cells of the normal liver were selected (clusters 4, 8, 11, 13, and 18) and reprocessed by the workflow described above to further identify epithelial and tumor subpopulations. A cluster of tumor cells expressing neuroendocrine markers, which was described in Song et al.[21], was specific to only one patient. Therefore, we did not focus on this cluster and excluded it from this analysis. In addition, because some of the identified clusters contained low-quality cells, contaminated by ambient RNA or cell fragments, high-quality clusters were selected (clusters 2, 3, 5, 6, and 8) and reprocessed once more by the same workflow, creating the final subset comprising 4 clusters.

For the primary tumors from PT9 and 13 and the organoid lines, raw reads were aligned using Cell Ranger (v6.1.1 and v7.0.1). The Seurat package standard workflow was employed as described before. Low-quality cells and doublets were filtered out from the primary cells and organoids separately based on the number of unique genes measured and the percentage of mitochondrial genes. After filtering, data was processed using the NormalizeData, ScaleData, and FindVariableFeatures functions for the RNA assay, and SCTransform to create an SCT assay. Dimensionality reduction was performed on the SCT assay using RunPCA, and UMAP coordinates were calculated using RunUMAP. Confounding genes were removed from the VariableFeatures as described above. Two separate clusters of organoid sample 135 were identified and labeled 135F$_2$ and 135E. All organoid idents were subsetted to 289 cells each. Dot plots and heatmaps were made using DoHeatmap or DotPlot, or the pheatmap package (v1.0.12), after markers were calculated using the FindAllMarkers or FindMarkers commands, using the RNA assay. The hierarchical clustering plot was made using BuildClusterTree. InferCNV was run using standard settings, with hepatocytes from the tissue object as reference[77]. pySCENIC (v0.11.2) was run on a high-performance cluster with standard settings, using a singularity file obtained from the Aerts lab[31]. After obtaining the AUC scores, Seurat was used to calculate differentially active regulons, using FindAllMarkers with logfc.threshold = 0.005. Gene set enrichment analysis was run using the fgsea package (v1.24.0), using h.all.v7.4.symbols.gmt.

For the reanalysis of the snRNA-seq tissue from Hirsch et al.[17], data was accessed through EGA (accession number EGAS00001005108). Cells were filtered as in the original paper and processed like the other datasets. The main tumor cluster (annotated in the original paper as containing cells from all three of their signatures) was subsetted, while a smaller cluster with possibly mesenchymal cells was omitted from our analysis. SCENIC was run as with the other datasets.

For the transcriptomic signature analysis, gene lists for different hepatoblastoma transcriptomic classifications were compiled from existing literature and compared with top significantly differentially expressed genes between our fetal and embryonal tumor populations (adjusted $p$-value < 0.05, greatest log2FC values) (gene lists are shown in Supplementary Data 1). A refined gene list of the bulk RNA-seq signatures from Hirsch et al.[17] was kindly provided by the authors. The AddModuleScore function from Seurat was used to calculate module scores for each gene set using log-normalized gene expression values. A correlation matrix between gene signatures and epithelial tumor clusters identified within this study was calculated based on the analysis described in Qin et al.[78] The module scores were z-scored to allow cross-signature comparison. Using the Corrplot (v0.92) R package, Pearson correlations were computed between the scores on all cells of the epithelial tumor clusters and then visualized as a correlation heatmap, grouped via complete linkage hierarchical clustering, only showing significant correlations (conf.level = 0.95). Signature scores were also visualized using FeaturePlots. An Upset plot was generated using the ComplexUpset (v1.3.3) R package. General WNT genes were curated from all WNT signaling pathway collections in GOBP, KEGG, PID, and Hallmarks (MSigDB v2023.2Hs), in addition to manually accrued WNT genes (Supplementary Data 3).

## Spatial data analysis

Raw FASTQ files and histology images were processed using Space Ranger (v1.2.2). Each sample was normalized individually using the SCTransform function of the Seurat R package with default parameters, except method = "glmGamPoi" to improve the speed and return.only.var.genes = FALSE. Clustering was performed using FindNeighbors and FindClusters. Clusters were annotated based on marker gene expression, differential gene expression using FindAllMarkers, and tissue histology. Clusters with similar gene expression profiles were combined. All samples were then merged with the merge function of the Seurat R package with default parameters.

## Organoid multiome analysis

Raw reads were aligned using Cell Ranger. Joint analysis was performed using the Seurat and Signac (v1.10.0) packages[79,80], following standard workflow unless specified otherwise. Cells were filtered using the following settings: "nCount_ATAC < 40000 & nCount_ATAC > 100 & nCount_RNA < 8000 & nCount_RNA > 300 & percent.mt < 2 & TSS.enrichment > 3 & nucleosome_signal < 2". For processing of the RNA assay, again SCTransform was used, mitochondrial and cell cycle (correlated) genes were removed from the VariableFeatures as before, and principal component analysis was performed using the RunPCA command. For the ATAC assay, peak calling was performed using macs2 (v2.2.9.1). Dimensionality reduction was then performed by running RunTFIDF, FindTopFeatures, and RunSVD. Next, multimodal analysis of both assays was performed, running FindMultiModalNeighbors with reduction.list = list("pca", "lsi") and dims.list = list(1:50, 2:50); RunUMAP with nn.name = "weighted.nn" and reduction.name = "wnn.umap"; and FindClusters with graph.name = "wsnn", algorithm = 3, and resolution = 0.2. Different lines were separated based on SNP-based demultiplexing and marker expression. For downstream analysis, all lines were downsampled to 400 cells each. For motif analysis, ChromVar (v1.26.0) was run after adding the human JASPAR2020 motifs. Differentially enriched motifs were calculated using wilcoxauc and plotted.

## Immunofluorescence staining

Whole pieces of tissues were fixed in 10% Neutral Buffered Formalin for 1 h to overnight, depending on the size of the specimen. Fixed tissues were processed using the Excelsior™ AS Tissue Processor (Thermo Fisher), following a standard protocol, and embedded in paraffin blocks. Organoids were released from the BME using dispase, washed, and fixed for 1 h in 10% Neutral Buffered Formalin. Fixed organoids were sequentially dehydrated through gradient alcohol and butanol, then paraffinized. The FFPE blocks were sectioned into 4 μm thick sections that were placed on SuperFrost Plus slides (Thermo Fisher). Sections were deparaffinized and rehydrated by immersing the slides in 3 changes of xylene followed by a series of decreasing concentrations of ethanol and rinsing in distilled water. Heat-induced antigen retrieval was performed by placing the slides in citrate/tris buffer and boiling for 20 min using a double boiler method. For IF staining, tissue sections were permeabilized in 0.3% Triton X-100 in PBS (PBS-Tx) for 5-10 min. Next, a blocking buffer (5% normal donkey serum [Jackson ImmunoResearch] in 0.1% PBS-Tx) was applied for 1 h at room temperature (RT). Tissue sections were incubated with primary antibodies (see Supplementary Table 5) diluted in blocking buffer for 1 h at RT or at 4 °C overnight and washed with 3 changes of 0.1% PBS-Tx. Secondary antibodies were diluted in PBS and applied for 1 hour at RT. All secondary antibodies were raised in donkey and conjugated to Alexa Fluor dyes (488, 555, and 647) (Thermo Fisher). Sections were washed with changes of PBS and counterstained with DAPI (Sigma-Aldrich) at 5 μg/ml for 5 min at RT. Slides were mounted in 80% glycerol in PBS and a #1.5 coverslip. Images were acquired with 20X dry and 40X oil immersion objectives on a Leica DMi8 or Thunder widefield microscope equipped with 4 LED light sources (DAPI, FITC, RHOD, and Cy5). Acquired images were adjusted for brightness and pseudo-colored using FIJI software. Signal quantification after nuclear segmentation was performed using CellProfiler, using standard settings.

## High-throughput drug screening

Organoids were recovered from their BME matrix by dispase incubation, followed by three washing steps with cold PBS containing 5% FBS. Organoids were then resuspended in organoid medium supplemented with 5% BME and plated in Corning 384 well microplates (Sigma-Aldrich, CLS3764) at ± 50 organoids per well, using a Multidrop Combi Reagent Dispenser (Thermo Fisher). The next day, the cell viabilities of one plate were measured by a Spectramax i3x plate reader using the CellTiter-Glo 3D Cell Viability Assay (CTG3D), and drugs were added to the other plates, using an Echo 550 liquid handler. We used the PMC library (an in-house developed library with > 200 drugs specific for pediatric cancer). Each drug was tested in duplicate and in 6 different concentrations (0.1 nM to 10 μM in increments of factor 10). After 120 h, the viabilities of all wells were measured using CTG3D. Graphs were then fitted, and area under the curve scores (AUCs) were calculated using R. A volcano plot was generated using the Wilcoxon signed-rank test, and the average AUC values for the two sub-groups were based on clustering. Z-scores against the pediatric tumor reference cohort were calculated using $(x - \mu) / \sigma$, where x represents the AUC of each drug in hepatoblastoma organoids, and $\mu$ and $\sigma$ are the mean and standard deviation of the AUC values for that drug in other pediatric tumors, respectively. Drugs with IC50 values higher than 10 mM were excluded from the analysis.

## Growth factor dependency experiments

Organoids were cultured for at least two weeks in BME in full medium with or without EFG or FGF10. Cell viability was measured according to the CCK-8 assay protocol (Medchem Express). Cells were incubated for 1 h at 37 °C. Optical density values were measured using the ClarioStar plate reader (BMG Labtech) at 450 nm. All samples were measured in technical duplicates.

## Erdafitinib sensitivity in FGF10 depleted culture medium

Organoids were cultured for at least two weeks in BME in full medium with or without FGF10. Next, organoids were cultured for 5 days in a medium (full medium with or without FGF10) containing different concentrations of erdafitinib, ranging from 1 nM – 10 μM, at equal concentrations of DMSO (0.1%). Cell viability was measured according to the CCK-8 assay protocol (Medchem Express). Cells were incubated for 1 h at 37 °C. Optical density values were measured using the ClarioStar plate reader (BMG Labtech) at 450 nm. All samples were measured in technical duplicates.

## Sanger sequencing

Genomic DNA was extracted using the DNeasy Blood & Tissue Kit (Qiagen) and *CTNNB1* exon 3 was amplified using GoTaq® G2 Flexi DNA Polymerase (Promega) according to the manufacturer's protocol. The PCR product and sequencing primers were sent to Macrogen Europe to perform Sanger sequencing. The sequences of the PCR and sequencing primers are shown in Supplementary Table 6.

## RNA isolation and qRT-PCR

Total RNA was extracted using the RNeasy Mini Kit (Qiagen) and reverse transcribed using the GoScript™ Reverse Transcriptase (Promega) according to the manufacturer's protocol. Quantitative RT-PCR was performed with GoTaq® qPCR and RT-qPCR Systems (Promega) on a CFX384 Real-time System (Bio-Rad). Relative target gene expression levels were calculated using the delta-delta CT method.

The primer sequences are shown in Supplementary Table 6.

## Western blot

Cells were lysed in a homemade lysis buffer. After quantification using the BCA protein Assay Kit (Thermo Fisher), protein samples were separated by sodium dodecyl sulfate-polyacrylamide gel electrophoresis (SDS-PAGE). These were then transferred onto polyvinylidene difluoride (PVDF) membranes (Millipore). Membranes were blocked in 5% BSA (Thermo Fisher) for 1 h at room temperature and incubated with antibodies against β-catenin (1:1,000; 8480 Cell Signaling Tech) and β-actin (1:3,000; 60008-1-Ig Proteintech), followed by incubation with IRDye 800CW goat anti-rabbit IgG secondary antibody (1:5,000; LI-COR) and IRDye 680RD goat anti-mouse IgG secondary antibody (1:10,000; LI-COR). Immunoreactive proteins were subsequently visualized using the Odyssey CLx Infrared Imaging System.

## *CTNNB1* mutation analysis

Sequencing analysis of tumor biopsy materials was performed in the diagnostic setting using the institute's standardized analysis pipelines. Single nucleotide variants and small indels in the *CTNNB1* gene (ENST00000349496.11/ENSP00000344456.5) were assessed using whole-exome and/or whole-genome sequencing. Exon 3 deletions were assessed based on bulk RNA-sequencing analysis after realignment of the BAM files using STAR aligner (v2.7.8a).

## Statistics and reproducibility

Sample sizes and statistical methods used are mentioned in the respective methods section and figure legends. No statistical method was used to predetermine sample size. No data were excluded from the analyses. The experiments were not randomized. The investigators were not blinded to allocation during experiments and outcome assessment.

## Data availability

The raw sequencing data generated in this study have been deposited to European Genome-Phenome Archive (EGA) under accession code EGAS50000000561. Due to patient privacy concerns, access to the sequencing data is managed by the Data Access Committee (DAC) of the Princess Maxima Center. All researchers can request access by submitting a project proposal to the DAC (biobank@prinsesmaximacentrum.nl). Requests are typically reviewed within approximately two weeks. The duration of access will be determined by the DAC. In addition, we used data from GSE186975[21] [https://www.ncbi.nlm.nih.gov/geo/query/acc.cgi?acc=GSE186975] and EGAS00001005108[17] [https://ega-archive.org/studies/EGAS00001005108]. Gene lists for different hepatoblastoma transcriptomic classifications were compiled from existing literature[14,17,18,21,30] and are shown in Supp. Data 1. Source data are provided in this paper.

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

## Acknowledgements

We would like to extend our gratitude to the Máxima Comprehensive Childhood Cancer (M4C) for Liver Tumor team, which includes Martine van Grotel, Kathelijne Kraal, Kees van de Ven, Lideke van der Steeg, Anneloes Bohte, Annemieke Littooij, Maarten Smits, Marijn Scheijde-Vermeulen, Liset Lansaat and Charlotte Aart. We would also like to extend our gratitude to the Princess Máxima Center Single-Cell Genomics Facility (Tito Candelli, Aleksandra Balwierz), Big Data Core and Kemmeren group (Ianthe van Belzen, Jayne Hehir-Kwa), the Stunnenberg group (Cristian Ruiz Moreno, Zhijun Yu) and the Van Heesch group (Jip van Dinter). Research reported in this publication was supported by Oncode Accelerator, a Dutch National Growth Fund project under grant number NGFOP2201, and by Kinderen Kankervrij (KiKa) Project 425. We are grateful for the generous donations from the public through the Princess Máxima Center Foundation (Kus van Kiki). The Princess Máxima Center Single-Cell Genomics Facility is supported by KiKa.

## Author contributions

Conceptualization: W.C.P. and J.Z. Methodology: W.C.P., T.A.K., Y.L., S.A.S., and L.J.K. Investigation: T.A.K., Y.L., S.A.S., L.J.K., F.R., M.v.d.W., V.A., and S.E. Software, analysis pipeline: T.A.K., S.A.S., P.L., W.L.M., T.M., and J.D. Visualization: T.A.K., S.A.S., and Y.L. Resources: W.C.P., H.C., J.Z., H.G.S., T.M., R.R.d.K., R.H.d.K., V.E.d.M, E.D., M.C.v.d.H., F.W.d.F., D.M., K.K., and J.J.M. Supervision: W.C.P., H.C., and H.G.S. Funding acquisition: W.C.P., J.Z., and H.G.S. Writing: W.C.P., T.A.K., and S.A.S. with inputs from all authors.

## Competing interests

H.C. is currently head of pharma Research Early Development (pRED) at Roche and is an inventor on several patents related to organoid technology. His full disclosure is given at www.uu.nl/staff/JCClevers. The remaining authors declare no competing interests.

## Additional information

¹Princess Máxima Center for Pediatric Oncology, Heidelberglaan 25, Utrecht, the Netherlands. ²Department of Hepatobiliary Surgery, Xiamen Hospital of Traditional Chinese Medicine, Beijing University of Chinese Medicine, Xiamen, China. ³Oncode Institute, Utrecht, the Netherlands. ⁴Department of Precision Medicine, University of Campania Luigi Vanvitelli, Vico L. De Crecchio 7, Naples, Italy. ⁵Department of Pediatric Hematology and Oncology, University Children's Hospital Münster, Albert-Schweitzer-Campus 1, Münster, Germany. ⁶Department of Pathology and Medical Biology, University Medical Center Groningen, Groningen, The Netherlands. ⁷Department of Surgery, Section of Hepatobiliary Surgery and Liver Transplantation, University of Groningen, University Medical Center Groningen, Groningen, the Netherlands. ⁸Department of Pathology, University Medical Center Utrecht, Heidelberglaan 100, Utrecht, the Netherlands. ⁹Hubrecht Institute, Royal Netherlands Academy of Arts and Sciences and University Medical Center, Utrecht, the Netherlands. ¹⁰Present address: Pharma, Research and Early Development (pRED) of F. Hoffmann-La Roche Ltd, Basel, Switzerland. ¹¹These authors contributed equally: Thomas A. Kluiver, Yuyan Lu. ✉e-mail: w.c.peng@prinsesmaximacentrum.nl

