## [Peer Review File · Nature Communications]

Reviewers' Comments:

Reviewer #1:

Remarks to the Author:

The manuscript by Kluiver et al. in line with other studies identified 2 distinct populations in human hepatoblastoma defined as fetal- and embryonal-like. These populations were validated using complementary methods in different cohort. Finally, the authors developed PDO from hepatoblastoma patients and performed a drug screening to define novel putative target. The article is interesting; however the novelty is limited.

Major comments

- As overall comment the authors should use a supplementary figure to illustrate the analysis performed on which samples were done (and the number of samples and cells used for each analysis).
- In the first paragraph of the results the authors claimed to focus on tumor cells alone in the re-analysis of data from previously published studies. TME has been demonstrated to be crucial in tumor development and in drug response. This reviewer would like to know the rationale behind this decision. This decision may affect the definition of the "signature" the authors defined across the manuscript.
- In the re-analysis of the previously published scRNA-seq the authors used only the data published by Song et al. and did not include the data from the manuscript by Huang et al. To increase the statistical power and the number of the cells the authors should also include the cells from the other manuscript and perform comparative (between the cohort) and overall (using both dataset) analyses. The results from these new analyses need to be integrated in the manuscript and used for the subsequent experiment performed.
- The authors in the second paragraph of the results denote fetal-like cells with an expression profile described previously (hepatic signature and Hepatic I and II). For clarity the authors should use the signature described in the other manuscript and apply to their dataset and show the results in a figure or supplementary item. This approach needs to be also used with the newly defined WNT-high expression profile/ embryonal like (described elsewhere as progenitor signature)
- In the Figure 1C there is not cut-off of significance, the authors need to define which of the different they describe are significant.
- The authors claim that cholangiocytes were enriched for specific regulons (Fig 1E) unfortunately the ones mentioned in the text are absent from the figure. The authors need to add a table in the supplementary data with all the information about the SCENIC results and revised the figure with the data described in the results.
- The second paragraph of the results for this reviewer it looks a circular argument given that the data used are the same, it is expected that cells with high expression of given genes should have a related TF more active and vice versa. The authors should use different datasets to cross validate the results. Moreover, some of these findings should be validated (for examples via IHC or similar techniques).
- ST on a different cohort of samples seems to largely recapitulate what observed using scRNAseq with a large heterogeneity. As mentioned in the first comment the authors should also use the other cell type (TME) in their analysis and using the ST for cross validation, also a ligand receptor analysis would be interesting. Furthermore, the authors observed a greater inter-tumor heterogeneity of fetal- and embryonal-like profiles in PT13, compared to the other 3 samples.

Could these differences be attributed to the tumor characteristic of the patient or are they related to the pre-treated nature of the sample? This point should be elaborated/discussed.

- The authors claimed “We observed separate regions of HNF4A+ cells and LEF1+ tumor cells, but also LEF1+ cells that were interspersed with HNF4A+ cells, as illustrated in PT13. Of note, islands of LEF1+ cells separated by stromal cells from surrounding HNF4A+ cells were also observed in post-chemotherapy samples (PT9, 17) (Figure 3).” Considering the good quality images, it would be worth to conduct a correlation analysis between staining (by a quantification of the markers and signal) and tumor samples this reviewer would expect a strong relation between staining and phenotype. Additionally, an analysis of nuclear b-catenin in relation to LEF1 and the state of b-catenin mutation could be interesting (relate to S4)

- For the scRNAseq analysis of the PDO the authors need to use the same signature used to discriminate the samples: embryonal like and fetal like, this will be a direct comparison with the samples, introducing a new classification is confusing. Of note this reviewer do not expect that the PDO will match exactly the tissues because the reasons already mentioned in the text.

- The authors should perform a full IHC and H&E characterization of the PDO and the original tissues.

- The authors should explain how the drugs for the screening were selected and which drug were present (supp table). Based on the previous findings (HDACi)?. Furthermore, it would be of interest to know if among the drug selected there were some WNT inhibitors and how they performed.

Minor comment

- Figure 1D is mentioned in the text before 1C, please correct.
- General comment: the authors in the drug screening experiments often omit numbers, % and ratios in the results. This might help the reader to already understand how much the two populations of hepatocytes are different.
- Figure S4 needs to be revised. 1) The labelling is not clear for each panel for both single staining and the overlay. Some samples (such as the Normal and PT10) seems to lack the DAPI staining. In addition, PT17 and PT15 the far-right image are not the same of the respective ones (left panel).

Reviewer #2:

Remarks to the Author:

Kluiver, Peng and collaborators performed comprehensive analyses of hepatoblastomas and tumor-derived organoids using single-cell RNA-seq, spatial transcriptomics, single-cell ATAC-seq and high-throughput drug profiling.

1- The authors identified two subclasses of tumor cells termed ‘fetal-like’ and WNT-high ‘embryonal-like’. This classification should be carefully compared to the previous HB transcriptomic classifications published by Cairo et al, Nagae et al, and Hirsch et al. Identification of similarities and differences will be important to mention.

2- A differential level of β -catenin activity was first described in C1 < C2 classification by Cairo et al., and similar results were obtained by Hirsch et al with the level of differentiation with Hepatocytic < liver progenitor. Could you please compare your results with the previously published one and tone down the novelty in the text.

3- Does the transcriptomic classification change during cell culture?

4- Sensitivity to FGFR inhibitors related to high FGFR expression is very interesting. Does the media of culture contain growth factors activating FGFRs? What happens if the media is depleted in such growth factor regarding the sensitivity to FGFR inhibitors? Does the level of FGFRs expression

- change if you compare primary tumors and organoids at early and late stage of culture?
- 5- Could you please expand on the mechanism at the origin of FGFR/EGFR and HDAC over-expression? Is it concerted expression in the same cells?
 - 6- More data are necessary to characterize better the expression level in tumors and organoids of FGFRs, EGFR and HDAC proteins. Cellular localization of these targets could be essential to interpret to predict response to inhibitors.
 - 7- Several figures are difficult to read because of the small lettering.
 - 8- How sequencing data will be accessible?

Reviewer #3:

Remarks to the Author:

In this new study by Kluiver, Peng and colleagues, the heterogeneity of liver cancer is explored through molecular profiling of tumors and organoid derivatives. One of the most important findings of this study is to identify and molecularly define two basic types of hepatic cancer cell: fetal-like and embryonal-like. The authors carry out an extensive genomic and transcriptomic analysis to identify the regulatory networks enriched in each subtype which informs on their drug-screening efforts to identify distinct vulnerabilities – vulnerabilities they connect selectively to EGFR and FGFR signal transduction. The molecular and drug sensitivity information is of high interest and value to both basic and clinical scientists interested in tumor heterogeneity, not just for hepatoblastoma, but for additional solid tumor types. The bioinformatic analysis are extensive and informative and of equal high value. However, tumor heterogeneity is an important, fundamental issue and the remaining questions and concerns about the current manuscript are centered on this issue:

1. Tumor heterogeneity exists as two types: Intra-tumor heterogeneity where single cells and/or clusters of cells of different cellular phenotypes co-develop within a tumor, versus Inter-tumor heterogeneity which refers to tumors comprised by one cancer cell phenotype or another. Here the authors provide examples of both types and it is therefore not clear how hepatoblastoma develops in general. For example, in Figure 1, of the nine patient scRNAseq datasets, most cells from tumors appear to be of the fetal type, with two tumors (? – it is difficult to tell with the resolution provided) comprising mainly embryonic-type cancer cells. This appears to imply that there is largely Inter-tumor heterogeneity, where individual tumors develop from one cancer cell subtype or the other. In Figure 2, three tumor samples are analyzed spatially. Two of these samples appear to be comprised of only the fetal-type of cancer cell, but the third sample is demarcated as a mixed embryonic-fetal sample. Overall, it is difficult to discern from the way that the data is displayed, and with the small number of tumors analyzed, just how heterogeneous hepatoblastoma is with regard to these two newly-identified subtypes, and what type of heterogeneity is common: Inter-? or Intra-?. This distinction is important because the authors end their study with a potentially useful drug-sensitivity survey, and they show that organoid cultures from tumors are homogeneous and individually distinct (Figure 4B). If hepatoblastomas are largely of one cancer subtype or the other, then this drug survey is directly –clinically relevant. On the other hand, if these two subtypes more often exist together Intra-tumorally, then treatment with one of the other might enrich for one subtype and/or trigger switching between subtypes. If the authors have data to inform on this crucial point, it would enhance the importance of this study.

2. Figure 3 (and Figure S4): there is a beautiful staining study of HNF4A and LEF1 in nine patient tumor samples and the authors conclude that there is mutually exclusive staining, making these two transcription factors biomarkers for the fetal and embryonic subtype. For the reader, mutual exclusivity is difficult to discern with the colors chosen to depict staining, and there are no statistics to support the conclusion. Additional analysis of this staining is needed to support the visual survey. For example, a histogram of pixel intensity for each biomarker on a cell-by-cell basis, or some other equivalent analysis. The question is whether these tumor cells display any mixed embryonic/fetal potential before and/or after chemotherapy. Strictly mutually exclusive

staining implies that there is no subtype switching between these two cancer types.

3. The authors derive organoids from tumors and perform an extensive molecular analysis. These are valuable data, but they need to be placed in the context of what the organoid culture actually represents. Overall, the organoids appear to be quite homogenous versions of their tumors and furthermore, each organoid culture is quite distinct in UMAP space from all the other organoids (Figure 4B). What is the origin of the homogeneity in organoid derivation and the distinctive clustering in UMAP space? How might that influence the subsequent assignment of a tumor organoid to Hepatic-High, Hepatic-Intermediate, or Wnt-High?

4. Several of the staining and ATAC-seq/Transcription Factor data provided seem to show that the fetal-Hepatoblastoma tumor subtype has additional heterogeneity. For example, Figure 5B and 5C (tumor sample 8F) is not uniformly expressing HNF4A, and has a distinct "enriched motif" profile, a unique profile seemingly stable to the organoid stage (Figure S5A, S5B: 8F, 96F, 17F tumor samples). Are there two fetal subtypes that vary based on HNF4A?

5. In the discussion, the authors propose that LEF1 and HNF4A be used as biomarkers in combination with existing markers to distinguish tumor samples and assign a hepatoblastoma subtype. The authors do not provide a proposed biomarker set, but given the extensive analysis presented, it would enhance the potential impact of the study if they could end by proposing a refined set of biomarkers for each subtype.

6. Minor Comments:

a. The heat map shown in Fig S3B depicts "low quality cells" that appear to this reviewer to be immune cells based on the markers ... could these low quality cells be B lymphocytes?

b. Cells designated "low quality" in Fig S4C have the same nFeatures (aka # of genes expressed) as the other clusters, but in this patient the "low quality cells" look stromal-like. Both for Fig S3B and S4C – assigning cells as Low Quality, rather than cell type is perplexing.

7. The discovery of differential drug sensitivity to EGFR and FGFR signaling inhibitors are significant and important, but there are a couple of perplexing differential responses. Lapatinib or Neratinib as shown in Fig S6E are joint EGFR/HER2 inhibitors and they are not very active (while the EGFR/HER2 inhibitors Afatinib and Sapitinib are selective). Can the authors address why Lapatinib and Neratinib are not as effective? Neratinib binds irreversibly as an inhibitor.

Reviewer #1 - PDOs, drug screening (Remarks to the Author):

The manuscript by Kluiver et al. in line with other studies identified 2 distinct populations in human hepatoblastoma defined as fetal- and embryonal-like. These populations were validated using complementary methods in different cohort. Finally, the authors developed PDO from hepatoblastoma patients and performed a drug screening to define novel putative target. The article is interesting; however the novelty is limited.

Response 1: We are thankful for the reviewer's insightful suggestions and appreciate the opportunity to emphasize the unique contributions of our study. This is the first study to provide a detailed characterization of a large hepatoblastoma organoid cohort across clinical stages and to identify specific inhibitors targeting hepatoblastoma. Moreover, we have introduced two transcription factors, HNF4A and LEF1, which, to our knowledge, are novel in their application for distinctly classifying tumor subtypes.

Major comments

1. As overall comment the authors should use a supplementary figure to illustrate the analysis performed on which samples were done (and the number of samples and cells used for each analysis).

Response 1.1: Thank you for the suggestion. We have added a schematic diagram to illustrate the general workflow and number of samples analyzed with each technique (Figure S1a). The number of cells per tumor analyzed in scRNA-seq is now added to Figure 1a and 2f.

2. In the first paragraph of the results the authors claimed to focus on tumor cells alone in the re-analysis of data from previously published studies. TME has been demonstrated to be crucial in tumor development and in drug response. This reviewer would like to know the rationale behind this decision. This decision may affect the definition of the "signature" the authors defined across the manuscript.

Response 1.2: We agree with the reviewer that the tumor microenvironment (TME) is important in cancer progression and drug response. In the original publication by Song *et al.* (Nature Communications, 2022), non-tumor clusters were described, with limited differences observed compared to their matching cell types from adjacent normal liver samples, except for: (i) the absence of MARCO expression, a marker for Kupffer cells (*i.e.*, liver resident macrophages), in tumor-associated macrophages and, (ii) the presence of 'erythroid progenitor cells' in tumor samples. The population of erythroid cells was also recently described by Wang *et al.* (Cell Reports Medicine, 2023). In parallel, we have performed a detailed characterization of the hepatoblastoma tumor immune microenvironment by antibody multiplexing using the imaging mass cytometry techniques. These findings have been described in a separate manuscript (Krijgsman and Kraaier *et al.*, bioRxiv, 2023).

The primary aim of this manuscript was the characterization of tumor organoids, through RNA-seq, to understand the molecular basis underlying their response to drug treatment. This effort has led us to identify HNF4A⁺ fetal and LEF1⁺ embryonal tumor components. To validate these findings in tissue samples, we revisited the single-cell RNA-seq data of previously published hepatoblastomas. Our reanalysis allowed us to recognize an embryonal signature in tumor tissues, that had not been detected in the initial study. This analysis further supports the validity of the tumor organoid signatures we observed.

3. In the re-analysis of the previously published scRNA-seq the authors used only the data published by Song at al. and did not include the data from the manuscript by Huang et "al. To increase the statistical power and the number of the cells the authors should also include the cells from the other manuscript and preform comparative (between the cohort) and overall (using both dataset) analyses.

The results from these new analyses need to be integrated in the manuscript and used for the subsequent experiment performed.

Response 1.3: We agree that including the Huang *et al.* (Hepatology, 2023) dataset would increase the sample size and statistical significance. We have made several attempts to obtain their data through the online database and through email contact with the authors but received no responses. We also contacted the editor of the journal (in July and again in October 2023) but the issue has still not been resolved.

To address this limitation, we have performed scRNA-seq analysis on two tumor samples from the PMC and analyzed a previously published single nucleus RNA sequencing (snRNA-seq) dataset (Hirsch *et al.*, Cancer Discovery, 2021). We identified both fetal and embryonal tumor cells across the three samples (Figure 1e-g). The tumor subtype-specific markers identified in the PMC samples and Hirsch's sample showed extensive overlap with our described signatures (Figure S1g). In addition, we performed gene regulatory network analysis on the datasets and identified high activation of HNF4A and LEF1 regulons respectively (Figure 1e,f). (**see also Response 2.1**)

4. The authors in the second paragraph of the results denote fetal-like cells with an expression profile described previously (hepatic signature and Hepatic I and II). For clarity the authors should use the signature described in the other manuscript and apply to their dataset and show the results in a figure or supplementary item. This approach needs to be also used with the newly defined WNT-high expression profile/ embryonal like (described elsewhere as progenitor signature).

Response 1.4: We agree with the reviewer suggestions to include a comparison to the hepatoblastoma signatures described in the original paper by Song *et al.* We have therefore plotted their signatures on our dataset, showing high expression of Hepatoblast I and Hepatoblast II in our fetal tumor cells (Figure S2a). Especially the genes of Hepatoblast I signature showed a high degree of overlap with our fetal tumor signature (Figure S2b). In addition, all three hepatic signatures show strong correlations with the other hepatic hepatoblastoma signatures from previous bulk RNA-seq studies (Figure S2d). Song's Hepatoblast I and Hepatoblast II signatures are derived in different tumor samples and likely illustrate patient-specific variations in gene expression profiles. None of the signature described by Song *et al.* could be linked to our embryonal-like signature, however enrichment of our embryonal-like tumor genes was observed in the Neuroendocrine and DCN-high tumor signatures, some overlapping WNT markers and mesenchymal markers (Figure S2b).

In addition, to the single-cell tumor signatures described by Song *et al.*, we have added comparison of our signatures to single-cell transcriptomic signatures described by Wu *et al.* (Biorxiv, 2023) and bulk RNA-seq signatures (Cairo *et al.*, Cancer Cell, 2008; Nagae *et al.*, Nature Communications, 2021; Hirsch *et al.* in Figure S2c,d), as requested by Reviewer 2 (**related to Response 2.1**).

5. In the Figure 1C there is not cut-off of significancy, the authors need to define which of the different they describe are significant.

Response 1.5: All of the gene sets shown in Figure 1d are significantly enriched. We have added the cut-off value ($p < 0.05$) to the figure and specified in the figure legend that adjusted *P*-values (< 0.05) were used for significance.

6. The authors claim that cholangiocytes were enriched for specific regulons (Fig 1E) unfortunately the ones mentioned in the text are absent for from the figure. The authors need to add a table in the supplementary data with all the information about the SCENIC results and revised the figure with the data the described in the results.

Response 1.6: Thank you for pointing out this inconsistency. These regulons were shown in a heatmap with the top 20 differentially active transcription factor regulons for each of the epithelial tumor and normal clusters in the Supplementary Figures of the previous manuscript. This figure has been moved to Figure S1d. The original Figure 1E has been moved to Figure S1f. We have added ONECUT1 and HNF1B to the dot plot. SOX9 was inadvertently added in the previous version and is now replaced by SOX4, another biliary transcription factor (Poncy *et al.*, Developmental Biology, 2015).

7. The second paragraph of the results for this reviewer it looks a circular argument given that the data used are the same, it is expected that cells with high expression of given genes should have a related TF more active and vice versa. The authors should use different datasets to cross validate the results. Moreover, some of these findings should be validated (for examples via IHC or similar techniques).

Response 1.7:

- i. We acknowledge that regulon scores should correspond to gene programs as revealed by RNA-seq analysis. Gene regulatory network (GRN) analysis provides a robust method for detecting key regulon activities based on the expression patterns of downstream target genes. We utilized GRN analysis to identify key transcription factors for the different subtypes. This approach is particularly effective in cases where transcription factors may have low RNA transcript levels yet remain active, as indicated by their regulatory influence on gene expression. Furthermore, we observed that GRN analysis eliminated patient-specific heterogeneity, based on clustering of organoid samples SCENIC UMAP (Figure 5b) versus RNA UMAP (Figure S5c).
- ii. We agree that the validation of our findings across different datasets would reinforce our conclusions. To this end, we expanded our analysis to include sc/snRNA-seq and GRN assessments on two additional datasets—one from the PMC cohort and another from Hirsch *et al.* Consistent with our original findings, we also identified fetal and embryonal tumor signatures in these datasets. Specifically, we observed one cluster with high activity in hepatic regulons, including HNF4A, and another cluster with high activity in WNT-related regulons, including LEF1 (Figure 1e,f).
- iii. Furthermore, we validated the expression of HNF4A/LEF1 through immunofluorescence in a separate cohort of tumor tissues. In the revised manuscript we have added two additional tissues, totaling 13 tumor samples and one adjacent normal liver sample, which are all shown in Figure S4b (previous Figure 3). Representative stainings are shown in Figure 3a. In addition, we have performed quantitative analysis (Figure 3b), providing evidence for the almost mutually exclusive expression of these transcription factors in tumor cells (**related to Response 1.9**). We have further validated this near mutually exclusive expression pattern of HNF4A/LEF1 within organoids using IF staining (Figure S6b).

8. ST on a different cohort of samples seems to largely recapitulate what observed using scRNAseq with a large heterogeneity. As mentioned in the first comment the authors should also use the other cell type (TME) in their analysis and using the ST for cross validation, also a ligand receptor analysis would be interesting. Furthermore, the authors observed a greater inter-tumor heterogeneity of fetal- and embryonal-like profiles in PT13, compared to the other 3 samples. Could these differences be attributed to the tumor characteristic of the patient or are they related to the pre-treated nature of the sample? This point should be elaborated/discussed.

Response 1.8: We acknowledge the reviewer's suggestion regarding the inclusion of the TME in our analysis and the potential insights that a receptor-ligand interaction analysis could provide. However, the resolution of our spatial RNA sequencing data, with 55 μm diameter spots, does not currently allow for the detailed analysis necessary to investigate these interactions.

Generally, we observed large necrotic and fibrotic regions, and reduced heterogeneity in tumor histology in the post-chemotherapy samples, compared to pre-treated samples. This is also seen by IF in Figure 3 and S4b. We think that the high degree of heterogeneity observed in PT13 is likely due to its untreated nature. Additionally, in the IF stainings in Figure S4b we also noticed that PT13 displayed a high degree of heterogeneity, with a notable mix between fetal and embryonal cells when compared to other pre-treated tissues. This example also highlights the difficulty in deciphering tumor signatures based on bulk RNA-seq.

9. *The authors claimed “We observed separate regions of HNF4A+ cells and LEF1+ tumor cells, but also LEF1+ cells that were interspersed with HNF4A+ cells, as illustrated in PT13. Of note, islands of LEF1+ cells separated by stromal cells from surrounding HNF4A+ cells were also observed in post-chemotherapy samples (PT9, 17) (Figure 3).” Considering the good quality images, it would be worth to conduct a correlation analysis between staining (by a quantification of the markers and signal) and tumor samples this reviewer would expect a strong relation between staining and phenotype. Additionally, an analysis of nuclear β -catenin in relation to LEF1 and the state of b-catenin mutation could be interesting (relate to S4)*

Response 1.9: Thank you for these suggestions. We have performed a quantification of nuclear signal intensity (as described in the revised Methods section) and added the results to Figure 3b. Based on this analysis, LEF1 and HNF4A expression is indeed almost mutually exclusive. In addition, as also suggested by the reviewer, we performed co-staining analysis of β -catenin and LEF1, and found that LEF1⁺ cells, on average, have higher nuclear β -catenin signal than LEF1⁻ cells (Figure 3c and S4b).

Regarding the correlation between staining and phenotype, we have incorporated the histology annotation provided by the pathologist, dr. de Krijger into Table S1. In most of the tumors annotated as ‘mixed fetal-embryonal’ by the pathologist, we observed both HNF4A and LEF1 expression. In samples annotated as ‘predominantly fetal’ by the pathologist, we observed mostly HNF4A staining patterns, with only a few LEF1⁺ cells noted.

10. *For the scRNAseq analysis of the PDO the authors need to use the same signature used to discriminate the samples: embryonal like and fetal like, this will be a direct comparison with the samples, introducing a new classification is confusing. Of note this reviewer do not expect that the PDO will match exactly the tissues because the reasons already mentioned in the text.*

Response 1.10: We appreciate that introducing a new classification is confusing. To be more consistent, we have renamed the organoid clusters ‘embryonal’ (previous “Hepatic low, WNT high”), ‘fetal-I’ (previous “Hepatic intermediate”) and ‘fetal-II’ (previous “Hepatic high”). In addition, we derived embryonal and fetal signatures from the primary tumor analysis and plotted these in our organoid models (Figure 4d). This confirms that the embryonal and fetal-II organoids recapitulate the different transcriptomic tumor groups.

11. *The authors should perform a full IHC and H&E characterization of the PDO and the original tissues.*

Response 1.11: We have added H&E stainings (Figure S5b) and HNF4A/LEF1 IF stainings (Figure S6b) for all organoids to the supplemental figures of the revised manuscript. The original tissues were reviewed by pathologists and characterized using the standard diagnostics IHC stainings for hepatoblastoma. Information on the most predominant histology of each tissue has been added to Supplementary Table 1.

12. *The authors should explain how the drugs for the screening were selected and which drug were present (supp table). Based on the previous findings (HDACi)? Furthermore, it would be of interest to know if among the drug selected there were some WNT inhibitors and how they performed.*

Response 1.12: We utilized an in-house curated drug library containing over 200 drugs, including more than 70 that are FDA-approved and target major pathways implicated in cancer. The use of this library is supported by previous studies conducted at our center (for example Calandrini *et al.*, Nat Comm. 2020; Vernooij *et al.*, Mol. Canc. Ther. 2021; Calandrini *et al.*, Cell Rep. 2021; Meister *et al.*, EMBO Mol. Med. 2022). A comprehensive list of all the drugs used and their targets has been added as a supplementary table (Table S4). Unfortunately, this library does not include WNT inhibitors. Most available WNT inhibitors, such as Porcupine inhibitors, target upstream mechanisms of β -catenin, which are likely ineffective in hepatoblastoma due to *CTNNB1* mutations. Additionally, drugs that directly target β -catenin's DNA-binding capabilities are scarce and none are currently approved for clinical use.

Minor comment

- *Figure 1D is mentioned in the text before 1C, please correct.*

Response 1.13: This is now corrected. Thank you for pointing out this discrepancy.

- *General comment: the authors in the drug screening experiments often omit numbers, % and ratios in the results. This might help the reader to already understand how much the two populations of hepatocytes are different.*

Response 1.14: To summarize the most important similarities and differences in drug responses between fetal and embryonal organoids, we have added a table to Figure 6b showing the average viability of each molecular subgroup at the highest dose for a selection of compounds. A more extensive table, containing these 8 drugs plus all EGFR compounds (additional 4 compounds), shows the average IC50 and average viability at the highest dose for each subgroup (Figure S7f).

- *Figure S4 needs to be revised. 1) The labelling is not clear for each panel for both single staining and the overlay. Some samples (such as the Normal and PT10) seems to lack the DAPI staining. In addition, PT17 and PT15 the far-right image are not the same of the respective ones (left panel).*

Response 1.15: We apologize for this omission of DAPI staining, which was performed but not shown in the previous version. We have now provided the individual and overlaid stainings in Figure S5b and improved the layout to increase clarity. In the new Figure 3a, we have selected novel regions for the LEF1-HN4A and β -catenin staining to better match the same regions between the different sections. As the images shown were cropped from a large area of tissues, we did our best to match to the same tissue regions between consecutive sections for each tissue (which can be several sections apart).

Reviewer #2 - Hepatoblastoma, scRNA-seq, ST (Remarks to the Author):

Kluiver, Peng and collaborators performed comprehensive analyses of hepatoblastomas and tumor-derived organoids using single-cell RNA-seq, spatial transcriptomics, single-cell ATAC-seq and high-throughput drug profiling.

1- The authors identified two subclasses of tumor cells termed 'fetal-like' and WNT-high 'embryonal-like'. This classification should be carefully compared to the previous HB transcriptomic classifications published by Cairo et al, Nagae et al, and Hirsch et al. Identification of similarities and differences will be important to mention.

Response 2.1: We thank the reviewer for this suggestion. We have added comparisons between our fetal and embryonal tumor clusters to the signatures published by Wu *et al.*, Cairo *et al.*, Nagae *et al.*, Hirsch *et al.*, and Song *et al.*

- i. In Figure 1h, we show the expression of Wu's Fetal and Embryonal gene signatures (obtained from histology-informed laser capture micro-dissected samples) in our fetal and embryonal subclusters. The expression of other published signatures are shown in our dataset in Figure S2a,c. Together this demonstrates high expression of the hepatic hepatoblastoma signatures (*i.e.*, Wu's Fetal, Cairo's C1, Nagae's Hepatocyte, Hirsch's Hepatocytic and Song's Hepatoblast I and II) in our fetal tumor subset and higher expression of the progenitor hepatoblastoma signatures (*i.e.*, Wu's Embryonal, Cairo's C2, Nagae's Proliferative, and Hirsch's Progenitor) in our embryonal tumor subset.
- ii. We also correlated the expression of these signatures in our tumor subsets, shown in Figure S2d, which confirms a strong correlation between the different hepatic hepatoblastoma signatures and between the progenitor hepatoblastoma signatures, while showing a negative correlation when comparing signatures from these two groups. Lower correlations, especially observed between the progenitor signatures from the different studies, can likely be contributed to the use of bulk tissue analysis (RNA-seq and microarrays) versus single-cell RNA-seq.
- iii. Furthermore, we compare the genes listed in the different signatures. This suggests a close resemblance between the transcriptomic classifications from Wu *et al.* and our signatures, based on the high number of overlapping genes that make up these signatures (Figure 1h). For the embryonal signatures this includes embryonal, WNT signaling and mesenchymal markers. For the fetal signatures is mainly comprises of liver metabolism markers. Note that only 50 markers in each group are provided by Wu *et al.*
- iv. For the remaining signatures we observed a limited number of overlapping genes (Figure S2e). For example, seven genes were shared between our fetal signature, Hirsch's hepatocytic and Nagae's hepatocyte signatures, all of which encode proteins integral to the metabolic functions of the liver. Between our embryonal signature and Hirsch's progenitor signature, we saw six overlapping genes, three of them are associated with embryonal development. Moreover, enrichment of WNT target gene expression has previously been described in hepatoblastomas versus normal adjacent liver by Cairo *et al.* and Sekiguchi *et al.* (Precision Oncology, 2020), and by Nagae *et al.* in their proliferative subtype. We specifically saw the enrichment of general WNT related genes in in our embryonal signature compared to the fetal signature (Figure S2f).

2- A differential level of β -catenin activity was first described in C1 < C2 classification by Cairo et al., and similar results were obtained by Hirsch et al with the level of differentiation with Hepatocytic < liver progenitor. Could you please compare your results with the previously published one and tone down the novelty in the text.

Response 2.2: Indeed, a correlation between tumor subtypes and β -catenin levels has been described previously by Cairo *et al.* and Hirsch *et al.*, based on bulk RNA-seq, which typically include a mixture of various tumor components. In this study, we used IF to illustrate the two distinct tumor components by HNF4A and LEF1. Moreover, we performed co-staining of LEF1 with β -catenin and found higher nuclear accumulation of β -catenin in LEF1⁺ than LEF1⁻ cells (**related to Response 1.9**).

In addition, we have adjusted the text to place our results in the context of the previous literature and highlighted the novelty of our work in the Results and Discussion sections.

3- Does the transcriptomic classification change during cell culture?

Response 2.3: We do not see the conversion of one organoid subtype to another during culture, based on fetal and embryonal tumor organoids determined by their morphology, gene expression profile, regulon profile, protein expression of HNF4A and LEF1, and/or drug screening profile. An overview of the different passages is shown in Figure S1a. In Figure S6b, we have performed staining of HNF4A and LEF1 in all organoids, at various passages between 2-10, and noted that each sample almost exclusively contains one molecular subtype.

4- Sensitivity to FGFR inhibitors related to high FGFR expression is very interesting. Does the media of culture contain growth factors activating FGFRs? What happens if the media is depleted in such growth factor regarding the sensitivity to FGFR inhibitors? Does the level of FGFRs expression change if you compare primary tumors and organoids at early and late stage of culture?

Response 2.4: We thank the reviewer for raising this question. The culture medium does indeed include FGF10. To investigate the effect of FGF10, we performed an experiment where tumor organoids were cultured without FGF10 for >2 weeks. The removal of FGF10 did not reduce the growth of tumor organoids, implying that FGF signaling within these organoids is most likely due to autocrine signaling (Figure S7d). In line with this, we observed the expression of FGF ligands within both tumor organoids and tumor cells from tissues (Figure 6j, S7c). We then analyzed the sensitivity of tumor organoids to FGFR inhibitor treatment when cultured with and without FGF (Figure S7g). We found that embryonal organoids, regardless of the presence of FGF10 in the culture medium, were susceptible to FGFR inhibitors. By contrast, fetal organoids did not show sensitivity to FGFR inhibitors in either condition. This indicates a subtype-specific response to FGFR inhibition, with embryonal organoids being distinctly responsive. Our experiments were performed in a cohort of organoids cultured in different passages (p2-11), in which we did not observe a correlation between passage number and FGFR expression nor passage number and sensitivity to FGFR inhibitors.

5- Could you please expand on the mechanism at the origin of FGFR/EGFR and HDAC over-expression? Is it concerted expression in the same cells?

i. EGFR expression is present in normal hepatocytes (Figure 1d) and EGFR signaling has been observed in hepatocytes during homeostasis and regeneration, such as following injury in vivo and cell culture in vitro. Using transgenic mice overexpression wild type β -catenin, the Monga group reported that EGFR is a WNT target in the liver (Tan *et al.* Gastroenterology, 2005). They also examined hepatoblastoma tissues by IHC and noted increased EGFR staining in tumors (7/10 samples). We also validated the expression of EGFR by IF, and observed co-expression of EGFR in HNF4A⁺ cells, and thus restricted expression of EGFR to fetal tumor cells (Figure 6h and S7i). This is consistent with high sensitivity of fetal tumor organoids to EGFR inhibitor, *i.e.*, Afatinib treatment at the highest dose resulted in an average viability in fetal organoids of 30% versus 60% in embryonal organoids (Figure 6c). This information for the 4 additional EGFR inhibitors in our drug screen is provided in Figure S7f.

- ii. We detected the expression of FGFR1/2 in the embryonal components of tumor tissues and organoids (Figure 1d and 6i), compared to fetal components. FGFR1 is upregulated by WNT pathway in murine liver with APC knockout, as shown previously (Benhamouche *et al.*, *Dev Cell*, 2006), consistent with strong WNT signaling in the embryonal tumors. In addition, FGFs, such as FGF8, play an important role in murine liver development (Gordillo *et al.* Development, 2015) and were broadly upregulated in our embryonal organoids (Figure 6j). This expression profile in embryonal organoids likely conferred broad sensitivity to pan-FGFR inhibitors, *.i.e.*, Erdafitinib treatment at the highest dose resulted in an average viability in fetal organoids of 67% versus 21% in embryonal organoids (Figure 6c, S7e,f). By contrast, FGFR4, an important regulator of bile acid metabolism in normal hepatocytes, is upregulated only in the fetal cells, but not the embryonal cells (Figure 6i).
- iii. The overexpression of HDACs in hepatoblastomas, compared to normal liver, has been described previously (Beck *et al.*, *Cancer Biol. Ther.*, 2016). HDACs have been shown to work together with β -catenin to regulate the expression landscape (Billin *et al.*, *Mol. Cell Biol.*, 2000; Rivas *et al.*, *Cell Mol. Gastroenterol. Hepatol.*, 2021). We think that HDAC1/2, which share high sequence homology, and functionality, are broadly upregulated in all tumor cells, given the near complete response of both fetal and embryonal organoids to romidepsin and panobinostat at sub-micromolar concentration.

Taken together, high EGFR expression is associated with fetal hepatoblastoma, while many components of the FGF signaling pathway are upregulated in embryonal hepatoblastoma. While HDAC1/2 are broadly expressed in hepatoblastoma.

6- More data are necessary to characterize better the expression level in tumors and organoids of FGFRs, EGFR and HDAC proteins. Cellular localization of these targets could be essential to interpret to predict response to inhibitors.

Response 2.6: In the previous version of the manuscript we had reported the mRNA expression of FGFs/FGFRs, EGFR and HDACs in tissue (Figure 1c, Figure S6) and organoids (Figure 6e,l and S7b,c). We have added violin plot of the expression of FGFs in organoids from our scRNA-seq results to the revised manuscript (Figure 6j), reflecting the increased expression in embryonal organoids compared to fetal organoids.

We performed immunofluorescence staining of EGFR on tumor tissues from four patients (Figure S7i). We observed expression of EGFR in HNF4A+ cells, consistent with RNA-seq data (Figure 1d). Unfortunately, we did not obtain good staining results with antibodies directed against FGFR and HDACs, which we have tested so far.

7- Several figures are difficult to read because of the small lettering.

Response 2.7: We appreciate the feedback and have revised the figures and legends throughout the manuscript to enhance their readability.

8- How sequencing data will be accessible?

Response 2.8: All processed data is deposited to GEO, with accession number GSE242139 code, and will be made public upon publication of the manuscript. A data availability statement has been added to the manuscript.

Reviewer #3-- Heterogeneity, ATACseq (Remarks to the Author):

In this new study by Kluiver, Peng and colleagues, the heterogeneity of liver cancer is explored through molecular profiling of tumors and organoid derivatives. One of the most important findings of this study is to identify and molecularly define two basic types of hepatic cancer cell: fetal-like and embryonal-like. The authors carry out an extensive genomic and transcriptomic analysis to identify the regulatory networks enriched in each subtype which informs on their drug-screening efforts to identify distinct vulnerabilities – vulnerabilities they connect selectively to EGFR and FGFR signal transduction. The molecular and drug sensitivity information is of high interest and value to both basic and clinical scientists interested in tumor heterogeneity, not just for hepatoblastoma, but for additional solid tumor types. The bioinformatic analysis are extensive and informative and of equal high value. However, tumor heterogeneity is an important, fundamental issue and the remaining questions and concerns about the current manuscript are centered on this issue:

Response 3: We thank the reviewer for their positive feedback and for understanding the relevance of our findings. We hope to have addressed their concerns about tumor heterogeneity in our response below.

1. Tumor heterogeneity exists as two types: Intra-tumor heterogeneity where single cells and/or clusters of cells of different cellular phenotypes co-develop within a tumor, versus Inter-tumor heterogeneity which refers to tumors comprised by one cancer cell phenotype or another. Here the authors provide examples of both types and it is therefore not clear how hepatoblastoma develops in general.

For example, in Figure 1, of the nine patient scRNAseq datasets, most cells from tumors appear to be of the fetal type, with two tumors (? – it is difficult to tell with the resolution provided) comprising mainly embryonic-type cancer cells. This appears to imply that there is largely Inter-tumor heterogeneity, where individual tumors develop from one cancer cell subtype or the other. In Figure 2, three tumor samples are analyzed spatially. Two of these samples appear to be comprised of only the fetal-type of cancer cell, but the third sample is demarcated as a mixed embryonic-fetal sample. Overall, it is difficult to discern from the way that the data is displayed, and with the small number of tumors analyzed, just how heterogeneous hepatoblastoma is with regard to these two newly-identified subtypes, and what type of heterogeneity is common: Inter-? or Intra-?.

This distinction is important because the authors end their study with a potentially useful drug-sensitivity survey, and they show that organoid cultures from tumors are homogeneous and individually distinct (Figure 4B). If hepatoblastomas are largely of one cancer subtype or the other, then this drug survey is directly clinically relevant. On the other hand, if these two subtypes more often exist together Intra-tumorally, then treatment with one of the other might enrich for one subtype and/or trigger switching between subtypes. If the authors have data to inform on this crucial point, it would enhance the importance of this study.

Response 3.1:

i. We appreciate the reviewer's feedback. To provide a clearer representation, we have added a table showing the number of cells/ST spots in fetal and embryonal clusters per patient, as observed in both scRNA-seq (Fig. 1a) and spatial RNA-seq (Fig. 2f) analyses. Indeed, it appears that tumors from specific patients contain predominantly one tumor subtype (*i.e.*, fetal or embryonal tumor cells). This observation may stem from the fact that the tumors analyzed in Song *et al.* were post-chemotherapy (except for one tumor), typically containing little viable tissue. Similarly in Figure 2, the spatial data generated by us were from chemotherapy treated patient, except for PT13. This pre-treatment tumor sample showed more viable tumor tissue and more heterogeneity in differential gene expression profiles between different regions (**related to response 1.8**). In addition, hepatoblastoma is a histologically heterogeneous disease, consisting of fetal, embryonal, and other components within each tumor (Ranganathan *et al.* Pediatric and Developmental Pathology, 2020). However, utilizing our fetal and embryonal tumor markers (IF for HNF4A and

LEF1 in Figure 3a and S4b), we were able to show the presence of intra-tumor heterogeneity not always clearly discernable on H&E stainings (Figure S4a).

- ii. Our drug screening data shed light on the varied responses of these subtypes to different treatments, underscoring the underlying biology and the dependency on specific signaling pathways. The analyzed tumor tissues (13 samples IF cohort) appeared to contain a mix of both molecular subtypes. As the reviewer pointed out, using a particular drug could potentially enrich for another tumor subtype. Although we currently lack data for this particular interaction, our findings emphasize the critical need for molecular characterization of the tumor prior to drug treatment. More importantly, we have also identified drugs, including HDAC inhibitors, that effectively target various tumor subtypes. These drugs are particularly effective in killing hepatoblastoma tumor cells, emphasizing their suitability based on our screening.

2. Figure 3 (and Figure S4): there is a beautiful staining study of HNF4A and LEF1 in nine patient tumor samples and the authors conclude that there is mutually exclusive staining, making these two transcription factors biomarkers for the fetal and embryonic subtype. For the reader, mutual exclusivity is difficult to discern with the colors chosen to depict staining, and there are no statistics to support the conclusion. Additional analysis of this staining is needed to support the visual survey. For example, a histogram of pixel intensity for each biomarker on a cell-by-cell basis, or some other equivalent analysis. The question is whether these tumor cells display any mixed embryonic/fetal potential before and/or after chemotherapy. Strictly mutually exclusive staining implies that there is no subtype switching between these two cancer types.

Response 3.2: We thank the reviewer for their compliment on our staining and their suggestion to perform quantification analysis. For better visualization, we have revised Figure 3a and S4b. In Figure 4b we added the individual images of DAPI, HNF4A, and LEF1 staining, in addition to the HNF4A/LEF1 overlay. Furthermore, to enhance clarity, the colors of the fetal (in blue) and embryonal cells (in orange) are now changed and consistent for these tumor subtypes throughout the manuscript.

We have added quantitative analysis as requested (**related to Response 1.9**), by nuclei segmentation and quantification of signal (pixel) intensity for the HNF4A and LEF1 (Figure 3c and S4c). Consistent with our observations, we did not observe co-staining of HNF4A and LEF1, in pre- and post-chemotherapy samples.

3. The authors derive organoids from tumors and perform an extensive molecular analysis. These are valuable data, but they need to be placed in the context of what the organoid culture actually represents. Overall, the organoids appear to be quite homogenous versions of their tumors and furthermore, each organoid culture is quite distinct in UMAP space from all the other organoids (Figure 4B). What is the origin of the homogeneity in organoid derivation and the distinctive clustering in UMAP space? How might that influence the subsequent assignment of a tumor organoid to Hepatic-High, Hepatic-Intermediate, or Wnt-High?

Response 3.3: In our analysis, the tumor organoids clustered according to sample origin on the UMAP, which we attribute to inter-tumoral heterogeneity. This pattern mirrors observations in other studies; for example, in the Song *et al.* dataset, tumor cells clustered by tumor origin, whereas non-tumor cells clustered by cell type. A similar pattern has been observed in other cancer types, such as melanoma (Tirosh *et al.*, Science, 2016). Notably, we opted not to perform batch correction on the organoid data to avoid potential issue of “over-correction” which could obscure meaningful biological variations between tumor samples. Instead, we were able to employ PCA analysis to classify the different tumor organoids.

Regarding the derivation of organoids, they typically represent either fetal or embryonal subtypes in culture, although from three patients we were able to obtain both components. This discrepancy might be linked to the specific tumor components present in the small tissue samples used for culture. Given the limited amount of starting material — often smaller than 1 mm² — it is challenging to sample a wide range of tumor regions for organoid culture. Additionally, the homogeneous gene expression profiles observed across the organoids could be influenced by the culture medium. As all organoids undergo several passages in the same medium, this environment may select for the proliferation of specific tumor clones. The fact that organoids from multiple patients contribute to different groups (Fetal-I, Fetal-II, Embryonal as shown in PCA analysis) reinforces the biological relevance of these findings. Their consistent response to drug treatment further supports this conclusion.

4. Several of the staining and ATAC-seq/Transcription Factor data provided seem to show that the fetal-Hepatoblastoma tumor subtype has additional heterogeneity. For example, Figure 5B and 5C (tumor sample 8F) is not uniformly expressing HNF4A, and has a distinct “enriched motif” profile, a unique profile seemingly stable to the organoid stage (Figure S5A, S5B: 8F, 96F, 17F tumor samples). Are there two fetal subtypes that vary based on HNF4A?

Response 3.4: We agree with the reviewer that our organoid data points towards (at least) two fetal subgroups, however we saw HNF4A expression in both subgroups. For clarity, in the revised manuscript we have renamed these groups to Fetal-I and Fetal-II. We have now performed HNF4A and LEF1 staining for all tumor organoid samples (Figure 6b). Using another HNF4A antibody that works better for organoid staining, we found that the Fetal-I (former hepatic intermediate) organoids were all positive for HNF4A. This organoid group had enrichment of additional and distinct motifs (Figure 5d), such as JUN/FOS, which are also upregulated in cholangiocytes (Figure S1f) and murine hepatocyte organoids (Peng *et al.* Cell, 2018), which indeed suggests additional heterogeneity within the fetal subgroups. Of note, for optimal growth, Fetal-I organoids needed to be cultured in reduced medium, except for 8F₁, highlighting the distinctive nature of Fetal-I compared to the other two subgroups of organoids. In addition, we were able to derive organoids from two post-chemotherapy liver resected samples, both of which fell within the Fetal-I group.

5. In the discussion, the authors propose that LEF1 and HNF4A be used as biomarkers in combination with existing markers to distinguish tumor samples and assign a hepatoblastoma subtype. The authors do not provide a proposed biomarker set, but given the extensive analysis presented, it would enhance the potential impact of the study if they could end by proposing a refined set of biomarkers for each subtype.

Response 3.5: We thank the reviewer for this suggestion. We have now added a diagram in Figure 7 illustrating the general features of the two tumor components, including their drug responses and highlighted a few markers for the different components. A more extensive list of differentially expressed genes is now provided in Supplementary Table 2.

6. Minor Comments:

a. The heat map shown in Fig S3B depicts “low quality cells” that appear to this reviewer to be immune cells based on the markers ... could these low quality cells be B lymphocytes?

b. Cells designated “low quality” in Fig S4C have the same nFeatures (aka # of genes expressed) as the other clusters, but in this patient the “low quality cells” look stromal-like. Both for Fig S3B and S4C – assigning cells as Low Quality, rather than cell type is perplexing.

Response 3.6: We thank the reviewer for pointing out these inconsistencies. Both these samples were chemo-treated tumors. We reviewed the histology of these samples (*i.e.*, the matching H&E stainings), and both the “stroma” and “low quality” regions contained fibrotic (necrotic) tissue with some cell dense

regions, which is possible immune infiltrate. In our previous analysis our parameters for quality control were likely too stringent to account for the combination of low cell density and low cell diversity in these regions. We have taken this into account in the revised manuscript.

- i. Re-annotation tumor PT16: although the spots previously annotated as “low quality” cluster, showed a lower number of features compared to the other clusters, we agree with the reviewer that these spots still contain valuable information. In addition, we noticed from the differentially expressed genes of the stroma cluster that (1) these genes consisted of immune markers (e.g., *CCL21*, *CCL19*, *NR4A1*, *CCL2*, *GPNMB*, *SRGN*) and (2) were also expressed in the “low quality” cluster. In the revised version (Figure S3e) we have therefore merged these two clusters. To highlight the presence of immune markers, we have indicated them in the new heatmap.
- ii. Re-annotation tumor PT14: as suggested by the reviewer we have merged the “stroma” and “low quality” clusters into a large stroma cluster in the revised manuscript, due to the expression of fibroblast-associated genes (e.g., *COL1A1*, *COL1A2*, *SPARC* and *AEBP1*) in these spots (Figure 3f).

7. The discovery of differential drug sensitivity to EGFR and FGFR signaling inhibitors are significant and important, but there are a couple of perplexing differential responses. Lapatinib or Neratinib as shown in Fig S6E are joint EGFR/HER2 inhibitors and they are not very active (while the EGFR/HER2 inhibitors Afatinib and Sapitinib are selective). Can the authors address why Lapatinib and Neratinib are not as effective? Neratinib binds irreversibly as an inhibitor.

Response 3.7: We have looked extensively into existing literature on the mechanisms of action of the different EGFR inhibitors included in our drug screening library. The IC50 values for EGFR differ widely per drug, with lower IC50 values for Afatinib, Sapitinib and Erlotinib (0.5 nM, 4 nM and 2 nM, respectively), compared to Lapatinib and Neratinib (10.2 nM and 92 nM, respectively). In addition, potential differences might also include solubility, drug stability, metabolism, and cellular uptake, which can influence the therapeutic range of a drug. Finally, when comparing the cell viability in fetal organoids with the reference cohort (Figure S7h), all five drugs seemed to be more effective in our fetal organoids.

Revision to text and figures.

In summary, we have made the following changes to the text and figures:

- We have revised the Introduction, Results, and Discussion to improve flow and clarity. Text referring to new data and analyses has been highlighted in grey.
- The methods section has been updated to include descriptions of new experiments.
- We have updated the literature references to relate our findings to previous studies.
- We have updated the funding sources section.
- We have also revised the layout of all main figures, supplementary figures and supplementary tables to increase clarity.
- We have added four new supplementary tables to support our analyses.

Figure 1.

1a. We have added a table with the number of cells per patient per tumor subgroup included in our analysis.

1e,f. We have validated our fetal and embryonal signatures by analyzing scRNA-seq of two PMC patients and an external snRNA-seq dataset (Hirsch *et al.*).

1h. We have included comparisons of our fetal and embryonal signatures to published transcriptomic hepatoblastoma signatures described by Wu *et al.*

Figure S1.

S1a. We added an overview of the samples we used in this study.
S1c,d. We have increased the size of the heatmaps to increase legibility.
S1f. The dot plot of SCENIC regulon activity scores was moved from the main figure (previous 1e). We have corrected an error in the labeling and included some additional markers.
S1g. We added heatmaps of the analysis of the dataset from Hirsch *et al.* the two PMC patients (related to Figure 1e,f).

Figure S2.

S2a-e. We have included comparisons of our fetal and embryonal signatures to published transcriptomic hepatoblastoma signatures.
S2f. We have added a bar plot of enrichment of general WNT genes.

Figure 2.

2f. We have added a small table of number of spots per patient for tumor region. We have also renamed “fetal-like” and “embryonal-like” to “fetal-enriched” and “embryonal-enriched”, respectively.

Figure S3.

S3a-h. We have combined the previous S2 and S3 into one large supplementary figure containing all additional panel on the spatial transcriptomics analysis.
S3e,f. We have performed re-analysis of the non-tumor regions of PT14 and PT16.

Figure 3.

We have added two additional tissues, totaling 13 tumor samples and one adjacent normal liver sample.

3a. Representative immunofluorescence stainings of LEF1-HNF4A and β -catenin are shown for different compositions of tissues.
3b. We added quantification analysis of LEF1-HNF4A co-staining.
3c. We added quantification analysis of LEF1- β -catenin co-staining.

Figure S4.

S4a. We have included H&E staining of each tissue.
S4b. We have moved the immunofluorescence stainings of LEF1-HNF4A of all tissues from the main figure to the supplementary figure. We have also added β -catenin stainings for the entire tissue cohort.
S4c. We have added quantification of nuclear HNF4A, LEF1 and β -catenin signals.

Figure 4.

We have expanded our scRNA-seq dataset with two new organoid models, 27F₁ and 28F₁. We have renamed the three organoid subgroups to, E, F1 and F2.
4c. We performed re-done of the GSEA to compare the three groups (instead of Fetal versus embryonal in the previous version)
4d. We added the expression of our fetal and embryonal signatures in our organoid samples.

Figure S5.

S5a. Phase-contrast pictures of the organoids were moved from the main figure (previous Figure 4a).
S5b. We added H&E stainings of all organoids.
S5c. UMAP was moved from the main figure (previous 4b).
S5d. We included a UMAP including early and late passage of 13F2.
S5e,f,g. We have increased the size of the heatmaps to increase legibility.

Figure 5.

5a. This panel was moved from Figure 4f in the previous version of the manuscript.

5b. We included a UMAP using the SCENIC data.

5d. For clarity we now visualize the top 10 markers (compared to the top 20 in the previous version of the manuscript)

5e. We have changed the layout to increase clarity of the immunofluorescence staining of the organoids.

Figure S6.

S6a. We have added a heatmap of differently expressed genes identified in the snRNA-seq data.

S6b. We have performed LEF1-HNF4A staining in our organoid lines.

S6c. We have performed β -catenin staining in our organoid lines.

Figure 6.

6b,g. We split the dose-response curves over two panels to increase clarity.

6c. We have added a table with the average viability per subgroup at the highest concentration for selected HDAC, FGFR and EGFR inhibitors.

6h. We have performed EFGR staining in hepatoblastoma tissues. Here we show a representative staining.

6j. We added the expression of FGFs from scRNA-seq data of the organoids.

Figure S7.

S7a. We have increased the size of the heatmaps to increase legibility.

S7d. We have showed organoid growth factor dependency for EGF in fetal organoids and we have showed that both fetal and embryonal organoids grow independently of FGF10.

S7f. We added a table with the average viability per subgroup at the highest concentration for all HDAC, FGFR and EGFR inhibitors.

S7g. We show FGF10 withdrawal from the culture medium does not affect sensitivity to FGFR inhibitors.

S7i. We have performed EFGR staining in hepatoblastoma tissues. Stainings for all tissues are shown.

Figure 7.

We have added a schematic overview of the major differences between embryonal and fetal hepatoblastoma cells, described in our study.

Table S1.

We have added two additional columns to Table S1 containing copy-neutral loss of heterozygosity (cnLOH) of chromosome 11p15.5 and predominant histology assessed by a pathologist (RRdK). We have also added two additional samples (PT28 and PT31).

Table S2.

We have added this table containing gene lists used in the new analyses depicted in Figure S2. This table contains the top 200 significant differentially expressed genes between our fetal and embryonal tumor cells and the gene signatures described in the literature (Wu *et al.*, Cairo *et al.*, Song *et al.*, Hirsch *et al.*, Nagae *et al.*),

Table S3.

We added this gene list of general WNT genes, used in the analysis depicted in Figure S2f.

Table S4.

We added this table containing all the compounds in our drug screening library with additional information, such as the drug's targets.

Reviewers' Comments:

Reviewer #1:

Remarks to the Author:

The manuscript has improved after the revision of the manuscript.

Reviewer #2:

Remarks to the Author:

I thank the authors for their answers to all my questions, and for adding new results in the manuscript, this is a huge work. I have nothing to add.

Reviewer #3:

Remarks to the Author:

Kluiver, Peng and colleagues have revised their manuscript to include additional analyses of datasets produced by other groups, and new scRNAseq and bulk RNAseq datasets from additional patient samples. While there are some differences in concordance, overall there appears to be robust alignment of these other tumor datasets with the Fetal I, II and Embryonic subtype annotations developed here. The authors have also provided an extensive set of expression and staining data for tumors and organoids as Supplemental Data. It is clear that LEF1 and HNF4A robustly demarcate separate tumor subtypes, with very little co-expression to suggest subtype switching. To this reviewer, the authors have done a great deal of work to address the reviewer's comments and the result is a mature study with important new findings both in the further characterization and clarification of tumor subtypes and in identifying drug sensitivities.

Minor Comments:

Figure 3C: In the legend or figure define the statistics for the three asterisks in the bar graph

Figure 7: Wnt regulons of LEF1, TCF7 and TCF7L1 are listed in the figure, however, the manuscript text describes TCF7L2 (not TCF7L1, which can be quite different in transcription regulation potential and activity). Please correct this.

Figure 7: Why is the FGF signaling not included as a marker? Given its emphasis in subtype classification and drug responses, the authors should consider adding this to the figure – perhaps in the "Enriched Pathways" category.

Figure S5f: The regulons of the organoids (heatmap) do not show HNF4A as enriched (and they show SOX9 as an enriched regulon). Clearly HNF4A is expressed in these organoids, but is regulon signature "weaker" in in vitro culture? This could hint at some plasticity in this subtype.

Reviewer #4:

Remarks to the Author:

Response to Reviewer's comments:

Figure 3C: In the legend or figure define the statistics for the three asterisks in the bar graph

- Thank you for pointing out this omission. This has now been added.

Figure 7: Wnt regulons of LEF1, TCF7 and TCF7L1 are listed in the figure, however, the manuscript text describes TCF7L2 (not TCF7L1, which can be quite different in transcription regulation potential and activity). Please correct this.

- Thank you for pointing out this error. It has been corrected to TCF7L2. We also added NR1H4 to the fetal group.

Figure 7: Why is the FGF signaling not included as a marker? Given its emphasis in subtype classification and drug responses, the authors should consider adding this to the figure – perhaps in the “Enriched Pathways” category.

- We have revised this as suggested.

Figure S5f: The regulons of the organoids (heatmap) do not show HNF4A as enriched (and they show SOX9 as an enriched regulon). Clearly HNF4A is expressed in these organoids, but is regulon signature “weaker” in in vitro culture? This could hint at some plasticity in this subtype.

- HNF4A is a differentially enriched regulon, but only the top 10 regulons per line are shown. HNF4A is shown on the heatmap if we increase the number of regulons. The violin plot also clearly shows the differences between groups.

Additional minor changes:

- The HDAC inhibitor IC50 values for organoid model 27F₁ were not included in the HDAC inhibitor sensitivity plot (Figure 6d). We have added this.
- qPCR expression data of *HDAC* and *FGFR* genes of organoid model 135F₂ was not included in figures 6e and j. We have added this.
- Minor adjustments in main figures for consistency and clarity: Fig. 1f (circles around embryonal and fetal clusters), Fig. 1g (added “etc.”), Fig. 5a (removed FXR from NR1H4 label).
- Minor adjustments in supplementary figures: Supp. Fig. 3a (better H&E staining of PT14 is now shown), Supp. Fig. 6c,f (consistently labelled organoid line 135 as 135F₂), Supp. Fig. 6d (removed statistical markers per organoid model as suggested, but added a comparison between fetal and embryonal groups).